# ERBench: An Entity-Relationship based Automatically Verifiable Hallucination Benchmark for Large Language Models

**Jio Oh**[*1]  **Soyeon Kim**[*1]  **Junseok Seo**[1]  **Jindong Wang**[2]  **Ruochen Xu**[3]

**Xing Xie**[2]  **Steven Euijong Whang**[†1]

[1]KAIST  [2]Microsoft Research Asia  [3]Microsoft Azure

## Abstract

Large language models (LLMs) have achieved unprecedented performances in various applications, yet evaluating them is still challenging. Existing benchmarks are either manually constructed or are automatic, but lack the ability to evaluate the thought process of LLMs with arbitrary complexity. We contend that *utilizing existing relational databases based on the entity-relationship (ER) model is a promising approach for constructing benchmarks* as they contain structured knowledge that can be used to question LLMs. Unlike knowledge graphs, which are also used to evaluate LLMs, relational databases have *integrity constraints* that can be used to better construct complex in-depth questions and verify answers: (1) *functional dependencies* can be used to pinpoint critical keywords that an LLM must know to properly answer a given question containing certain attribute values; and (2) *foreign key constraints* can be used to join relations and construct multi-hop questions, which can be arbitrarily long and used to debug intermediate answers. We thus propose ERBench, which uses these integrity constraints to convert any database into an LLM benchmark. ERBench supports continuous evaluation as databases change, multimodal questions, and various prompt engineering techniques. In our experiments, we construct LLM benchmarks using databases of multiple domains and make an extensive comparison of contemporary LLMs. We show how ER-Bench can properly evaluate any LLM by not only checking for answer correctness, but also effectively verifying the rationales by looking for the right keywords.

## 1 Introduction

Large Language Models (LLMs) [1, 2] have become prevalent and are increasingly popular in a wide range of applications, including natural language processing, chatbots, content generation, and information retrieval, to name a few. However, a fundamental issue of LLMs is hallucination [3–6], which refers to the phenomenon that LLMs generate fake, unverified, or non-existent information especially for knowledge-related and safety-critical applications. Hallucination remains one of the most severe issues that should be addressed, and we focus on factual hallucination.

To address factual hallucination, it is necessary to develop benchmarks that are comprehensive, intricate, automatically verifiable, and can be scaled efficiently. One approach is to construct manual benchmarks by human annotators [7–10], which are expensive and not scalable. Another approach is to automatically construct evaluation samples using knowledge graphs [11] by converting triples to simple factual questions or use existing QA datasets [12]. Although these benchmarks may scale as

---

[*]Equal contribution {harryoh99, purplehibird}@kaist.ac.kr. [†]Corresponding author ⟨swhang@kaist.ac.kr⟩

38th Conference on Neural Information Processing Systems (NeurIPS 2024) Track on Datasets and Benchmarks.

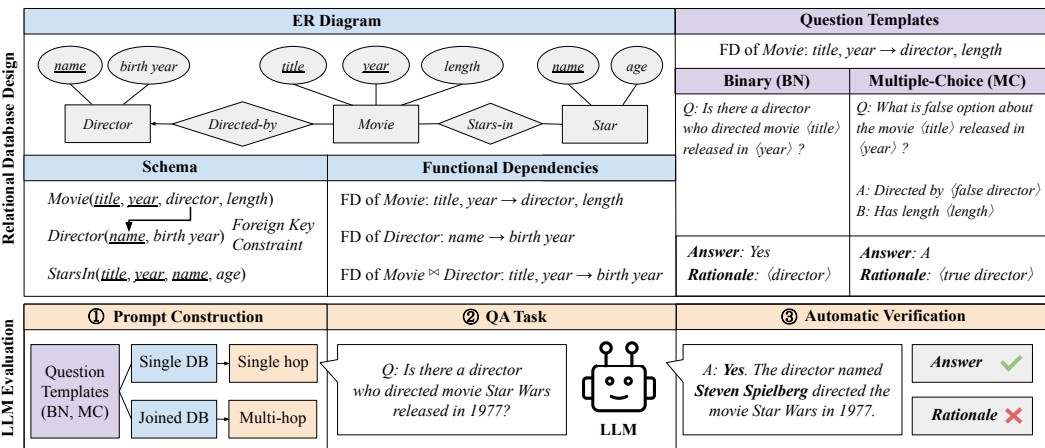

Figure 1: ERBench constructs questions from a relational database using its schema, records, and integrity constraints and automatically verifies the LLM responses.

the answers can be verified automatically and updated with more knowledge, their questions are still simplistic or unmodifiable, thus lacking the ability to evaluate on intricate tasks.

We contend that utilizing existing relational databases is a promising approach to construct a benchmark that has both merits. Until now, many LLM benchmarks have been constructed based on knowledge graphs. Although these benchmarks can scale, the main limitation is that the questions tend to be simplistic as they are based on triples. In comparison, relational databases contain structured data where they have schema information and follow the entity-relationship (ER) model, which supports various integrity constraints that make sure the data is well formed. A schema can be designed using traditional ER diagrams or more recent notions like UMLs. By using a database's schema, records, and integrity constraints, it is possible to construct arbitrarily-long multi-hop questions based on multiple relations that also have clear automatically-verifiable answers based on the integrity constraints. Using databases thus opens up opportunities to extensively evaluate LLMs on a vast amount of knowledge in a principled fashion.

In this paper, we propose **ERBench**, an LLM benchmark based on the ER model. ERBench supports complex questions and are automatically verifiable (see Fig. 1). The questions can be automatically constructed using ER diagrams. For example, if a movie's length is determined by its *title* and *year*, and the ER diagram shows the entity movie with three attributes *title*, *year*, and *length*, then one can ask an LLM *Does the movie titled Star Wars produced in 1977 run for more than 60 minutes?*. We use two popular integrity constraints – functional dependencies (FDs) and foreign key constraints (FKCs) to make the questions verifiable. FDs are used to infer an attribute's value based on other attribute values. In our example, if the FD *title*, *year* → *director*, *length* holds for movies, then the director and length of *Star Wars (1977)* are determined (*George Lucas* and *121 minutes*, respectively). We can construct both binary and multiple-choice questions asking for the inferred values. A foreign key is a set of attributes in one relation that refers to the primary key of another relation, which identifies records, and an FKC ensures that the foreign key values actually exist in the other relation. Using FKCs, ERBench can support questions with increasing complexity by generating multi-hop questions via joining multiple relations that have FKCs and inferring longer FDs that span them.

ERBench is also extensible in terms of data, modality, and prompting. First, ERBench can be easily updated as its underlying database changes and thus support continuous evaluation. Second, ERBench supports multimodal questions where one can replace attribute text values with other data types like images. Third, we can further diversify the questions using recent techniques like chain-of-thought[13], few-shot prompting[14], and knowledge augmentation[15–17]. ERBench can thus evaluate any improved LLM.

We conduct extensive experiments using 5 public databases and evaluate several popular LLMs: GPT-3.5[18], GPT-4[1], Llama2-70B-Chat[19], Gemini-Pro[2], Claude-3-Sonnet[20], and Mistral-7B-Instruct[21]. We perform comprehensive analyses in terms of answer and rationale accuracies and hallucination rates using single-hop, multi-hop, and multimodal questions and also perform prompt engineering and fine-tuning. We show how ERBench can effectively evaluate any LLM by

not only checking for answer correctness, but also effectively verifying their rationales by looking for the critical keywords that should be mentioned.

**Summary of Contributions.** (1) We propose ERBench, the first LLM benchmark to systematically utilize relational databases to construct complex questions where the model reasoning can be automatically verified. (2) We show how any database can be converted to a benchmark using its schema, records, and integrity constraints. (3) We extensively evaluate contemporary LLMs using ERBench and demonstrate how ERBench is effective and scalable, remaining relevant over time.

## 2 Preliminaries

**LLM Factual Hallucination**   We would like to evaluate LLMs in terms of their hallucination levels. We define hallucination as generated content that is nonsensical or unfaithful to the provided source content [22]. Hallucination may occur because the data sources are flawed in various ways or the data is utilized imperfectly and cannot be recalled properly [22]. In any case, we are interested in whether an LLM not only gives correct answers, but also has the correct thought process and thus consider hallucination on two levels: (1) The LLM gives a wrong answer. For example, if the question asks whether Firenze and Florence are the same city, the answer is *Yes* (Firenze is the Italian name of Florence), and *No* is considered as hallucination. (2) The LLM gives a correct answer, but with a wrong rationale. If the answer is *Yes*, but the rationale is that both cities are in the United States, then this is considered as hallucination as well. We note that recent hallucination benchmarks like Head-to-Tail [11] are good at evaluating (1), but are not designed to evaluate (2). Our key idea is to utilize relational databases to generate questions that can be used to evaluate both (1) and (2).

**Relational Databases**   We utilize relational databases, which are based on the ER model. A relation consists of a schema containing attributes and records containing the attribute values. When designing a schema, a typical approach is to start with an intuitive ER diagram or UML to determine which entities and relationships are needed and how they are connected with each other. For example, the movie ER diagram in Fig. 1 has three entity sets *Movie*, *Star*, and *Director*. In addition, there could be relationships, e.g., *Stars-in* between *Movie* and *Star* and *Directed-by* between *Movie* and *Director*. The resulting schema of the database could then be *Movie*(*title*, *year*, *director*, *length*), *StarsIn*(*name*, *age*), and *Director*(*name*, *birth year*).

The relations usually have integrity constraints. A *functional dependency* (FD) is a relationship between two sets of attributes $X$ and $Y$ where the $X$ values determine the $Y$ values. For example, Fig. 1 shows the FD *title, year → director, length* because a movie's title and year can be used to identify the movie, which in turn determines its director and length. Likewise, there is another FD *name → birth year* where the director name determines his or her birth year. We denote an FD as $X \rightarrow Y$ and use it to evaluate the LLM's rationale as we explain later. A *foreign key constraint* (FKC) is a set of attributes in one relation that refers to the primary key attributes in another relation, which is used to identify records. Fig. 1 shows that the *director* attribute of *Movie* is a foreign key to the *name* attribute of *Director*. Using a foreign key, we can also join two relations and construct FDs that span them. In Fig. 1, the FD *title, year → birth year* is a result of joining the *Movie* and *Director* relations and combining the first two *Movie* and *Director* FDs. This multi-relation FD construction enables us to construct questions with arbitrary complexity as we explain later.

The integrity constraints thus determine the correctness of records, and our idea is to utilize them to verify the LLM responses as well. A natural question to ask is whether the integrity constraints themselves are always correct. Since the integrity constraints are determined by the database owner, it is the owner's responsibility to determine if they should hold in general.

## 3 ERBench

We explain how ERBench utilizes FDs to construct two types of questions – binary and multiple-choice – and automatically verifies LLM responses. We then explain how complex multi-hop questions are constructed by joining relations with FKCs and expanding the FDs. Finally, ERBench can be extended to diverse data types, modalities, and prompt engineering.

### 3.1 Binary and Multiple-choice Question Construction using FDs

Given a relation $R$ and an FD $X \to Y$, we can specify the $R.X$ values and ask a question involving the $R.Y$ values. For example, using the database in Fig. 1, we can ask the question $q_1$: *For the movie with the title ⟨Harry Potter and the Philosopher's Stone⟩ produced in ⟨2001⟩, is the length larger than ⟨100⟩ minutes?*. Optionally, the question can be generated by an LLM using the ER diagram.

We can construct binary questions where the answers are *Yes* or *No*. We do allow an LLM to answer *Unsure* if it is not confident about its answers in order to significantly reduce hallucinations [11]. Hence, similar to Head-to-Tail [11], we use the prompt *Answer the following question in yes or no, and then explain why. Say unsure if you don't know and then explain why*. Note that we ask for further explanation to also verify the LLM's rationale. For $q_1$ above, the correct answer is *Yes* as the movie is 152 minutes long.

We can also construct multiple-choice questions where we provide several correct options and one incorrect option, which the LLM needs to figure out. The correct options can be generated using any FD. The incorrect option can be generated by choosing one FD $X \to Y$ where we know the correct $Y$ value and choosing a different $Y$ value from the relation. Optionally, to check whether the LLM is not just guessing, we can also add the option *None of the above* and make it the correct answer. If we extend $q_1$ above, a correct option would be *The movie length is 152 minutes*, while an incorrect one can be constructed by replacing the 152 minutes with another length in the relation.

### 3.2 Automatic Verification of LLM Responses

When verifying an LLM response, ERBench checks if both the answer and rationale are correct. The answer checking is straightforward where we check if the LLM selected the right binary or multiple-choice option. To check the rationale, we look for the inferred values of the FD applied on the current record. For example, let us assume the FD *released year, star, director → title*. If we ask the binary question $q_2$: *Is there a movie, released in ⟨2001⟩, starring ⟨Emma Watson⟩ where ⟨Chris Columbus⟩ is the director?*, we not only look for the answer *Yes*, but also the movie title *Harry Potter* within the rationale. The checking for multiple-choice questions is similar.

We may run into an entity resolution problem where the LLM mentions an entity that is essentially the same as the inferred one, but is written differently. In the above example, we may be looking for *Harry Potter*, but the LLM mentions *Harry J. Potter*, which contains the middle initial of *Harry Potter*. We perform entity resolution based on heuristics including conventional string matching. Another possible solution is to use an LLM itself for the matching. While ChatGPT has indeed been used to replace human judgement [11], we also believe this may give an unfair advantage to GPT compared to other LLMs and choose not to use this method.

### 3.3 Multi-hop Question Construction using FKCs

We can increase the complexity of a question by making it a multi-hop question [23, 24], which can be used to test whether an LLM can think in multiple steps. We use the straightforward extension of joining multiple relations to implement the multiple hops. The relations must have foreign key relationships so that the FDs can span the relations. Given two relations $R(X, Y)$ and $S(Y, Z)$ where $R$'s key is $X$, suppose there is a foreign key constraint from $R.Y$ to $S.Y$. For any tuple $e$ representing an entity in $R$, we can ask a question only using its $X$ and $Z$ values to see if the LLM knows the hidden $Y$ value. For example, by joining the *Movie* and *Director* relations in Fig. 1, we can construct the 2-hop question $q_3$: *Was the director who directed the movie ⟨Harry Potter and the Philosopher's Stone⟩ that was released in ⟨2001⟩ born in the ⟨1950s⟩?*. Here $e$ is the original *Harry Potter and the Philosopher's Stone* movie, and the hidden $Y$ value is *Chris Columbus*. If the LLM knows *Chris Columbus*, then it should be able to confirm that his birth year is *1958*, while giving *Yes* as an answer. Notice that the verification of multi-hop questions is exactly the same as for single-hop questions. Thus, we can construct arbitrarily-complex, but automatically-verifiable questions.

### 3.4 Extensions

ERBench is extensible in terms of data, modality, and prompting. ERBench supports continuous evaluation in the sense that if the underlying databases change, ERBench can be updated automatically by simply reflecting the record updates to the questions as well. As LLMs need to be evaluated

Table 1: FDs and binary questions of two datasets. See Sec. A.2 for examples of other datasets.

| Dataset | FD | Example Question |
|---|---|---|
| *Movie* | *released year, star, director* → *movie title* | *Is there a movie, released in* ⟨*2009*⟩*, starring* ⟨*CCH Pounder*⟩ *where* ⟨*James Cameron*⟩ *is the director?* |
| *Soccer* | *nationality, club, jersey number* → *player name (2019 year only)* | *Is there a soccer player from* ⟨*Portugal*⟩ *who played for* ⟨*Juventus*⟩ *with uniform number* ⟨*7*⟩ *in* ⟨*Juventus*⟩ *in 2019?* |

with newer data over time, it is important that the benchmark itself is also updatable. ERBench can also support multimodal data by making the underlying database multimodal. For example, we can replace a text attribute in a relation with an image and then pose the same question to the LLM. Finally, LLMs can use any prompt engineering techniques and still be evaluated with ERBench. We highlight the ones we use in our experiments: (1) Chain-of-thought [13] is a step-by-step approach of prompting and is more likely to give an LLM better context for answering the question; (2) Few-shot prompting [14] provides demonstrations to the LLM to give it more context before answering questions; and (3) Knowledge augmentation [15–17] utilizes a search engine to augment the generated answer with search results. Note that ERBench is orthogonal to whatever prompt engineering is used.

## 4 Experiments

We test the performances of LLMs on questions based on databases that are constructed from public data. We evaluate LLMs based on whether their answers and rationales are both correct.

**LLMs Compared.** We compare GPT-3.5 [18], GPT-4 [1], Llama2-70B-Chat [19], Gemini-Pro [2], Claude-3-Sonnet [20], and Mistral-7B-Instruct [21]. For multimodal LLMs, we evaluate GPT-4V [1] and Gemini-Pro-Vision [2]. We access the LLMs through Hugging Face, Microsoft Azure AI Studio APIs, the Google Gemini API, or Anthropic's API. To exclude randomness in the LLM responses, we set all the temperature parameters to zero.

**Datasets and Functional Dependencies.** We perform experiments on 5 datasets representing different domains and call them *Movie* [25], *Soccer* [26], *Airport* [27], *Music* [28], and *Book* [29] (see Sec. A.1 for details). We also use separate *Director*, *Club* and *Olympic* relations that are joined with the *Movie* and *Soccer* relations for multi-hop questioning. All the data we use are available on Kaggle or public Github repositories. Table 1 and Table 8 (in Sec. A.2) show the FDs we use to verify the LLM responses for binary and multiple-choice questions, respectively.

**Performance Measures.** We utilize existing LLM hallucination measures [11] and newly introduce two measures involving rationale evaluation. All four measures are useful for accurately analyzing LLM hallucinations (see Sec. B.1 for details).

- *Answer Accuracy* (**A**) [11]: Portion of LLM responses that are correct.
- *Rationale Accuracy* (**R**): Portion of responses whose rationales contain the FD-inferred values.
- *Answer-Rationale Accuracy* (**AR**): Portion of responses that are not only correct, but also contain FD-inferred values in their rationales.
- *Hallucination Rate* (**H**) [11]: Portion of responses that are incorrect, excluding those where LLMs admit uncertainty in their responses (e.g., *Unsure*). Specifically, $\mathbf{H} = 1 - \mathbf{A} - \mathbf{M}$, where $\mathbf{M}$ denotes the percentage of LLM responses that admit they cannot answer the given question (i.e., *missing rate*). A lower **H** value is better.

**Measuring Performance based on Internal Knowledge.** When measuring LLM performances, we also take into account their internal knowledge of the entities within the questions. That is, if an LLM is hallucinating on a question because it simply does not know the entity mentioned (e.g., the entity may not exist in its training data) we may want to skip that question for evaluation. We can assess an LLM's knowledge by directly prompting if it knows an entity (e.g., *Do you know about the movie "Harry Potter"?*; see more details in Sec. B.2). For our datasets, the numbers of known entities per LLM are shown in Sec. B.3. For a fair evaluation, we perform three types of evaluations: (1) each LLM is evaluated only with questions with entities that it has knowledge of; (2) all LLMs are evaluated with questions that all the LLMs have knowledge of; and (3) all LLMs are evaluated

Table 2: LLM performances using the binary basic (BN$_{(Y)}$), binary negated (BN$_{(N)}$), and multiple-choice (MC) questions on the 5 datasets. "n/a" means the LLM knows too few entities (less than 20) for the result to be meaningful; see Sec. B.3 for the # of entities known by each LLM. Lower **H** values are better, whereas higher **A**, **R**, and **AR** values are better. See Sec. B.4 for more results.

| Model | Metric | Movie BN$_{(Y)}$ | BN$_{(N)}$ | MC | Soccer BN$_{(Y)}$ | BN$_{(N)}$ | MC | Airport BN$_{(Y)}$ | BN$_{(N)}$ | MC | Music BN$_{(Y)}$ | BN$_{(N)}$ | MC | Book BN$_{(Y)}$ | BN$_{(N)}$ | MC |
|---|---|---|---|---|---|---|---|---|---|---|---|---|---|---|---|---|
| GPT-3.5 | **A** | **.85** | .06 | .97 | .35 | .00 | .83 | .13 | .00 | .71 | .58 | .20 | .91 | **.77** | .01 | **.55** |
| | **R** | .81 | .11 | .96 | .28 | .02 | .60 | .01 | .00 | .44 | .36 | .18 | .68 | .13 | .01 | **.55** |
| | **AR** | **.80** | .05 | .96 | .24 | .00 | .55 | .01 | .00 | .44 | .34 | .14 | .66 | .12 | .00 | .12 |
| | **H** (↓) | **.15** | .94 | .03 | .64 | 1.0 | .17 | .67 | .97 | .29 | .27 | .76 | .09 | .05 | .94 | .45 |
| GPT-4 | **A** | .65 | .51 | .97 | .47 | .19 | **.91** | .68 | .11 | **.96** | **.83** | .63 | **.97** | .41 | .02 | .48 |
| | **R** | .81 | .76 | .97 | **.70** | .49 | **.69** | **.23** | **.16** | **.90** | **.74** | **.56** | .87 | **.21** | .02 | .34 |
| | **AR** | .64 | .50 | .97 | **.41** | .17 | **.69** | **.20** | **.03** | **.90** | **.68** | **.54** | .86 | **.19** | .01 | **.34** |
| | **H** (↓) | .35 | .47 | .03 | .38 | .04 | **.02** | .32 | .80 | **.04** | .15 | .01 | **.01** | .08 | .01 | **.01** |
| Llama2 | **A** | .05 | **1.0** | .93 | .02 | **1.0** | n/a | .00 | **1.0** | n/a | .76 | **1.0** | .91 | .02 | **1.0** | n/a |
| | **R** | .53 | **.92** | .97 | .18 | **.62** | n/a | .00 | .02 | n/a | .31 | .29 | .71 | .02 | .05 | n/a |
| | **AR** | .05 | **.92** | .91 | .02 | **.62** | n/a | .00 | .02 | n/a | .29 | .29 | .67 | .00 | **.05** | n/a |
| | **H** (↓) | .95 | **.00** | .07 | .98 | **.00** | n/a | 1.0 | **.00** | n/a | .24 | **.00** | .09 | .98 | **.00** | n/a |
| Gemini-Pro | **A** | .28 | .00 | .92 | .01 | .00 | .89 | .00 | .00 | .76 | .51 | .02 | .89 | .00 | .00 | .47 |
| | **R** | .66 | .27 | .97 | .06 | .05 | .55 | .00 | .00 | .33 | .14 | .05 | .55 | .00 | .00 | .13 |
| | **AR** | .26 | .00 | .92 | .00 | .00 | .51 | .00 | .00 | .33 | .11 | .01 | .54 | .00 | .00 | .11 |
| | **H** (↓) | .40 | 1.0 | .08 | **.17** | .29 | .11 | **.00** | .42 | .24 | **.02** | .10 | .11 | **.01** | .01 | .53 |
| Claude-3 -Sonnet | **A** | .30 | .15 | **.99** | .21 | .01 | n/a | .52 | .01 | .91 | .46 | .38 | **.97** | .24 | .00 | n/a |
| | **R** | **.87** | .86 | **.99** | .36 | .02 | n/a | .08 | .07 | .80 | .30 | .23 | **.93** | .18 | **.06** | n/a |
| | **AR** | .29 | .15 | **.99** | .17 | .01 | n/a | .05 | .01 | .76 | .26 | .23 | **.90** | .14 | .00 | n/a |
| | **H** (↓) | .70 | .83 | **.01** | .30 | **.00** | n/a | **.00** | **.00** | .09 | .33 | .01 | .03 | .10 | **.00** | n/a |
| Mistral | **A** | .48 | **1.0** | .72 | **.54** | .99 | .44 | **.71** | **1.0** | .13 | .41 | .96 | .56 | .16 | .98 | .40 |
| | **R** | .54 | .66 | .75 | .10 | .18 | .21 | .00 | .00 | .09 | .03 | .04 | .37 | .01 | .01 | .04 |
| | **AR** | .31 | .66 | .64 | .06 | .18 | .19 | .00 | .00 | .05 | .03 | .04 | .14 | .00 | .01 | .03 |
| | **H** (↓) | .52 | **.00** | .28 | .46 | .01 | .56 | .29 | **.00** | .87 | .54 | **.00** | .44 | .83 | .02 | .60 |

with all questions, regardless of their knowledge. Due to space constraints, we only present results of (1), and the (2) and (3) results can be found in Sec. B.4.

## 4.1 Results for Single-hop Questions

Table 2 shows the LLM performances using single-hop binary and multiple-choice questions on the 5 datasets. We first explain how we construct the questions and then analyze the LLM performances.

**Binary Questions.** We use the basic questions in Table 1 and expect a *Yes* answer. In addition, we construct negated versions of these questions and expect a *No* answer. For example, we can negate a basic question for the *Movie* dataset in Table 1 as *Is it true that there are no movies released in ⟨2009⟩, starring ⟨CCH Pounder⟩ where ⟨James Cameron⟩ is the director?*. The reason we always negate a basic question instead of say changing one of its attribute values into an incorrect one is to still perform the FD-based verification. If the question cannot utilize FDs anymore, we can no longer just look for inferred values, but need to analyze the entire LLM response to verify the rationale. Table 2 shows that GPT-4 tends to have superior performances, especially in terms of **A** and **R**. Gemini-Pro and Claude-3-Sonnet have lower **A** and **R**, but also low **H**. All three LLMs along with GPT-3.5 have worse performances for the negated questions. Llama2 and Mistral, on the other hand, tend to give trivial *No* answers for most questions, which results in high performances on the negated questions, but low performances on the basic ones.

**Multiple-choice Questions.** For the multiple-choice questions, we generate 2–4 choices using the attributes on the right-hand side of the FDs, with one incorrect choice using the construction in Sec. 3.1. For each question, we generate three versions with rephrased choices (e.g., "born in US" can be rephrased to "birthplace is US" and "place of birth is US") and report the averaged LLM performance to account for prompt sensitivity (see Sec. A.2 for more details). Table 2 shows that most LLMs perform better on multiple-choice questions than on binary questions, with Claude-3-Sonnet showing a particularly notable improvement. GPT-3.5 and Gemini-Pro show similar performances, and GPT-4 often results in the best performances. Llama2 and Mistral perform well for the *Movie*

dataset among the 5 datasets. We also extend the multiple-choice questions with a *None of the above* option and show that the LLM performances tend to decrease as the proportion of this option increases (see more details in Sec. B.6).

Instead of trying to rank LLMs by performance, we would like to make observations from a benchmark perspective: (1) rationale accuracy (**R**) tends to be worse than answer accuracy (**A**) and (2) the LLM performances vary significantly across question types, even when using the same entities. We thus conclude that there is much room for improvement for LLM rationale and that LLM benchmarking should always involve diverse questions for a comprehensive analysis.

## 4.2 Rationale Verification Accuracy

We evaluate ERBench's effectiveness in evaluating an LLM's rationale. ERBench's main strategy is to utilize FDs to pinpoint critical keywords that must appear in an LLM's rationale. However, there are inevitable corner cases, where a rationale contains the right

Table 3: ERBench's manual verification accuracy.

|  | GPT-3.5 | GPT-4 | Llama2 | Gemini | Claude-3 | Mistral |
|---|---|---|---|---|---|---|
| Acc (%) | 97.4 | 94.4 | 92.8 | 94.8 | 96.8 | 97.0 |

keyword, but is incorrect. To see if these cases are frequent, we manually inspect 500 randomly-chosen responses per model for single-hop questions across all datasets and see if ERBench correctly evaluates the rationales. Table 3 shows that ERBench's correctness is higher than 95.5% on average and higher than 92% for any model. We also perform an error analysis of these results and larger-scale experiments by comparing ERBench with GPT-Judge [7] in Sec. B.5, which show similar results.

## 4.3 Results for Multi-hop Questions

For the multi-hop questions, we construct 2-hop and 3-hop questions for the *Movie* and *Soccer* datasets, respectively, using the construction in Sec. 3.3 (see more details in Sec. B.7). Since we are evaluating multiple steps of the LLM's reasoning, we also extend the **R** and **AR** measures as follows:

- **R**-ext: We compute the portion of rationales that occur in the $i^{th}$ hop that are correct, for each $i \in$ [1, . . ., total # hops]. We then take the average of these portions.

- **AR**-ext: We compute the **AR** value of answers and rationales that occur in the $i^{th}$ hop, for each $i \in$ [1, . . ., total # hops], in order to analyze any "snowball effect" [30] on how early hop reasoning errors affect the final accuracy.

Table 4 shows the LLM performances on the two datasets. Compared to the single-hop question results, most LLMs naturally have worse answer and rationale accuracies. Even if the answer accuracies are high, the rationale accuracies are low, underscoring the need to evaluate LLM rationales. Using Chain-of-Thought prompting [13] (denoted as "+ CoT") has mixed results where the demonstrations sometimes guide the LLMs to retrieve better knowledge about entities, but may also add unintended biases. Finally, the **AR**-ext results show that the **AR** performance does not decrease for more hops, which means that answering the early hops correctly is important. In Sec. B.8, we also show the importance of the correctness of initial hops to avoid any snowballing of incorrectness.

## 4.4 Results for Multimodal Questions

We explore the extensibility of ERBench by incorporating multimodality, specifically images, into GPT-4V and Gemini-Pro-Vision. Using the multimodal question construction in Sec. 3.4, we introduce multimodality in single-hop questions of the two datasets: *Movie* and *Soccer*. We replace each *title* attribute value with a movie poster image in *Movie* dataset and each *club* attribute value with a soccer club logo in *Soccer* dataset (see the actual prompts in Sec. C.1). The results are shown in Table 5. Overall, the integration of image modality tends to improve the performance compared to Table 2. The improvements are more pronounced

Table 5: LLM performances using multimodal questions on 2 datasets.

| Model | Metric | Movie | | | Soccer | | |
|---|---|---|---|---|---|---|---|
| | | BN(Y) | BN(N) | MC | BN(Y) | BN(N) | MC |
| GPT-4V | **A** | **.93** | **.95** | **.92** | .45 | **.45** | **.90** |
| | **R** | **.97** | **.97** | **.85** | **.40** | **.38** | **.67** |
| | **AR** | **.91** | **.93** | **.84** | **.38** | **.38** | **.64** |
| | **H** (↓) | **.06** | **.05** | **.06** | .07 | **.00** | **.03** |
| Gemini -Pro-V | **A** | .82 | .17 | .59 | **.67** | .01 | .68 |
| | **R** | .95 | .94 | .73 | .38 | .20 | .56 |
| | **AR** | .78 | .16 | .58 | .34 | .01 | .43 |
| | **H** (↓) | .18 | .83 | .41 | .31 | .97 | .32 |

when using Gemini-Pro-Vision. ERBench is thus also effective in evaluating multimodal models.

Table 4: LLM performances using the binary basic (BN(Y)) and binary negated (BN(N)) multi-hop questions w/wo CoT prompting on 2 datasets. We exclude Mistral as it knows too few entities (less than 20), resulting in mostly "n/a" values; see Sec. B.3 for the # of entities known by each LLM. For each question type, we mark the best performance in bold among all models w/wo CoT prompting.

| Model | Metric | Movie & Director | | | | Soccer & Olympic | | | |
| | | w/o CoT | | w/ CoT | | w/o CoT | | w/ CoT | |
| | | BN(Y) | BN(N) | BN(Y) | BN(N) | BN(Y) | BN(N) | BN(Y) | BN(N) |
|---|---|---|---|---|---|---|---|---|---|
| GPT-3.5 | A | .74 | .00 | .36 | .54 | .81 | .82 | .59 | .76 |
| | R-ext | .92 | .92 | .95 | .79 | .09 | .15 | .67 | .60 |
| | AR-ext | .73/.70 | .00/.00 | .35/.34 | .53/.50 | .01/.05/.03 | .00/.00/.00 | .47/.54/.54 | .28/.31/.31 |
| | H (↓) | .21 | .96 | .62 | .27 | .17 | .18 | .33 | .21 |
| GPT-4 | A | .59 | .40 | **.80** | .66 | .46 | .55 | .79 | .70 |
| | R-ext | **.98** | **.98** | **.98** | **.98** | .59 | .60 | **.80** | **.80** |
| | AR-ext | .59/.59 | .40/.39 | **.80/.79** | .66/.65 | .38/.42/.41 | .18/.19/.19 | .52/.55/.55 | **.68/.73/.73** |
| | H (↓) | .41 | .60 | .20 | .34 | .25 | .13 | .20 | .29 |
| Llama2 | A | .02 | **1.0** | .79 | .06 | **.88** | .12 | .84 | .33 |
| | R-ext | .95 | .95 | .97 | .95 | .25 | .30 | .47 | .56 |
| | AR-ext | .02/.02 | **.98/.92** | .78/.77 | .06/.05 | .00/.00/.00 | .44/.55/.44 | .12/.14/.13 | .45/.62/.62 |
| | H (↓) | .98 | **.00** | .21 | .94 | .12 | .88 | .15 | .65 |
| Gemini-Pro | A | .19 | .01 | .31 | .02 | .00 | .00 | .18 | .41 |
| | R-ext | .42 | .60 | .40 | .28 | .01 | .07 | .60 | .46 |
| | AR-ext | .19/.19 | .01/.01 | .31/.30 | .02/.01 | .00/.00/.00 | .00/.00/.00 | **.66/.73/.75** | .16/.15/.21 |
| | H (↓) | .02 | .33 | **.00** | .20 | **.00** | **.00** | .60 | .15 |
| Claude-3 -Sonnet | A | .44 | .57 | .56 | .02 | .57 | **1.0** | .21 | .14 |
| | R-ext | .95 | .95 | .96 | .95 | .58 | .26 | .39 | .27 |
| | AR-ext | .44/.42 | .57/.55 | .56/.53 | .02/.02 | .21/.23/.18 | .33/.34/.33 | .58/.60/.59 | .39/.40/.40 |
| | H (↓) | .50 | .37 | .44 | .97 | .20 | .18 | .22 | .14 |

## 4.5 Results on Prompt Engineering Methods

We further diversify our questions using prompt engineering techniques for a more extensive evaluation of LLMs. In Sec. 4.3 , we use chain-of-thought [13] techniques in multi-hop questions to encourage step-by-step reasoning and observe improved LLM performances (Table 4). In addition, we use few-shot prompting [14] in single-hop questions where we provide 8 (2–4) demonstrations before asking each binary (multiple-choice) question (see detailed demonstration prompts in Sec. C.2). Table 23 in Sec. B.9 shows that the demonstrations indeed improve LLM performances for both types of questions for most domains. Finally, we implement a simple version of knowledge augmentation [15] using the LangChain API [31], which adds a background knowledge passage of entities from Wikipedia before each prompt. Surprisingly, we often observe a degradation in LLM performance as the passage may actually mislead the LLM instead of help it; see Table 24 and Sec. B.10 for a more detailed analysis.

## 4.6 Results for Fine-tuning

We analyze how fine-tuning affects LLM performance using ERBench. We use GPT-3.5 and fine-tune it for 2 epochs on (1) 3,000 entities of the *Soccer* dataset and (2) the combination of 4 datasets – *Movie*, *Soccer*, *Music*, and *Book* – to check whether data from different distributions can improve LLM performance. We then observe how its performances on the 5 datasets change. We use similar questions as in Sec. C.2. As a result, Table 6 shows that fine-tuning is mostly helpful across all datasets, but there is still room for improvement. Interestingly, increasing the number of datasets for fine-tuning does not necessarily lead to an increase in performance compared to just fine-tuning on the *Soccer* dataset, although there is still a boost compared to not fine-tuning. Similar to Sec. 4.3, some prompts for fine-tuning occasionally add unintended biases to the model.

# 5 Related Work

There are many benchmarks that evaluate LLM responses, and we categorize them by what they evaluate and whether they scale. There are also LLM hallucination detection methods that are more focused on improving the LLMs instead of evaluating them, and we summarize them in Sec. C.

Table 6: GPT-3.5 performances on (1) only the *Soccer* dataset and (2) the combined dataset of *Movie*, *Soccer*, *Music*, and *Book*. See Sec. B.3 for the number of entities known by GPT-3.5 after fine-tuning. Lower **H** values are better, whereas higher **A**, **R**, and **AR** values are better.

| Model | Metric | Movie | | Soccer | | Airport | | Music | | Book | |
|---|---|---|---|---|---|---|---|---|---|---|---|
| | | BN(Y) | BN(N) | BN(Y) | BN(N) | BN(Y) | BN(N) | BN(Y) | BN(N) | BN(Y) | BN(N) |
| GPT-3.5 | **A** | .85 | .06 | .35 | .00 | .13 | .00 | .58 | .20 | .77 | .01 |
| | **R** | .81 | .11 | .28 | .02 | .01 | .00 | .36 | .18 | .13 | .01 |
| | **AR** | .80 | .05 | .24 | .00 | .01 | .00 | .34 | .14 | .12 | .00 |
| | **H** ($\downarrow$) | .15 | .94 | .64 | 1.0 | .67 | .97 | .27 | .76 | **.05** | .94 |
| GPT-3.5 + FT w/ Soccer | **A** | .88 | **.89** | .92 | .92 | **.94** | .91 | **.93** | **.93** | **.94** | .68 |
| | **R** | .93 | **.95** | .75 | .71 | **.09** | **.08** | **.61** | **.55** | .09 | .10 |
| | **AR** | .84 | **.86** | .70 | .66 | **.09** | **.08** | **.60** | **.54** | .09 | .09 |
| | **H** ($\downarrow$) | .12 | **.11** | .08 | .08 | **.06** | .09 | **.07** | **.07** | .06 | .32 |
| GPT-3.5 + FT w/ 4 Datasets | **A** | **.95** | .34 | **.97** | **.96** | .51 | **1.0** | .20 | .91 | .50 | **.95** |
| | **R** | **.98** | **.95** | **.81** | **.78** | .07 | .07 | .53 | .53 | **.17** | **.17** |
| | **AR** | **.93** | .34 | **.78** | **.75** | .05 | .07 | .16 | .49 | **.13** | **.17** |
| | **H** ($\downarrow$) | **.05** | .65 | **.03** | **.04** | .49 | **.00** | .80 | .09 | .50 | **.05** |

Many benchmarks evaluate factual knowledge of LLMs, which can be (1) general [7, 32, 10, 33–40, 8, 9, 41] or (2) specialized, where the questions are about medicine and healthcare [42, 43], certain languages [44, 45], financial tasks [46], or car systems [47]. In comparison, ERBench can be applied to any domain as long as its underlying database reflects the knowledge of that domain.

Another line of benchmarks evaluates specific functionalities or qualities of LLMs. Functionality evaluations assess LLM performance on tasks such as long text generation [48, 49], semantic role identification [50], knowledge location [51], report or knowledge generation [52–54], text summarization [55, 56], attributing answers [57], fact checking [58], and multitasking [59]. Quality evaluations, on the other hand, focus on aspects like consistency [60, 61] and reliability [62, 12, 63, 64] of LLM responses. ERBench aligns most closely with fact-checking and reliability, but the key contribution is the automatic evaluation of both model answers and rationales.

Recent scalable LLM benchmarks utilize existing QA datasets [12, 10] or knowledge graphs [11] to automatically generate questions at scale. For example, one can easily convert subject-predicate-object triples in a knowledge graph into a question that asks for the object. However, these approaches fail to verify the thought process of LLMs as they only check whether the final answers are correct. In comparison, ERBench is the first to utilize relational databases for LLM evaluation and can perform automatic rationale verification and systematic multi-hop question generation by leveraging database integrity constraints. We provide a more detailed comparison with benchmarks based on knowledge graphs in Sec. C.

# 6 Conclusion

We proposed ERBench, which pioneers a new direction of LLM benchmark construction by systematically converting any relational database based on the ER model to an LLM benchmark. ERBench starts from a schema and generates questions using an ER diagram and ensures that the LLM responses can be automatically verified using functional dependencies in a principled fashion. In addition, ERBench uses foreign key constraints to join relations and construct multi-hop questions, which can be arbitrarily complex and used to evaluate the intermediate answers of LLMs. Finally, ERBench is extensible in terms of data, modality, and prompt engineering. We generated our own LLM benchmark in 5 domains and performed comprehensive analyses in terms of answer and rationale accuracies and hallucination rates using single, multi-hop, and multimodal questions and also performed prompt engineering and fine-tuning. Overall, ERBench is effective in evaluating any LLM's thought process by pinpointing critical keywords.

**Societal Impact & Limitation**    ERBench can perform comprehensive evaluations of LLMs, which can help LLM users make informed choices. ERBench's limitation is that it only checks for critical keywords to verify LLM rationales. Although we demonstrated that this strategy is very effective, an interesting future work is to verify rationales based on their entire contents.

## Acknowledgments and Disclosure of Funding

This research was supported by the MSIT(Ministry of Science, ICT), Korea, under the Global Research Support Program in the Digital Field program)(RS-2024-00436680) supervised by the IITP(Institute for Information & Communications Technology Planning & Evaluation). This project is supported by Microsoft Research Asia and the resource is supported by the Microsoft Accelerating Foundation Models Research (AFMR) Program.

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

# A   Appendix – More Details on Datasets and Functional Dependencies

## A.1   More Details on Datasets

Continuing from Sec. 4, we provide more details on the 5 datasets utilized in our experiments.[*]

- *Movie* [25]: We use a relation with the attributes *movie title*, *released year*, *director*, *country of origin*, *genre* and *4 main stars*. For multi-hop questioning, we join this relation with a separate *Director* relation with the attributes *director name* and *birth year*. We generate the *Director* relation using the crawled data from Wikipedia.

- *Soccer* [26]: We use a relation with the attributes *player name*, *club*, *jersey number*, *nationality*, and *league*. For multi-hop questioning, we join this relation with a separate *Club* relation with the attributes *club name* and *located city*, and *Olympic* relation with the attributes *city name* and *hosted years*. Note that the *Olympic* relation encompasses information pertaining to the Summer Olympics; for the sake of brevity, we refer to it simply as *Olympic*. We generate the *Club* and *Olympic* relation using the crawled data from Wikipedia.

- *Airport* [27]: We use a relation with the attributes *airport name*, *shortcode*, *latitude*, *longitude*, *located city*, and *country code*.

- *Music* [28]: We use a relation with the attributes *music title*, *artist name*, *released year*, and *genre*.

- *Book* [29]: We use a relation with the attributes *book title*, *author*, *published date*, and *publisher name*.

We note that personal information included in the above datasets, such as the names of soccer players and movie stars, are all sourced from publicly available and reputable resources like official soccer league websites and online movie databases. Additionally, we have ensured that the datasets do not contain any offensive content.

## A.2   Functional Dependencies and Question Templates

Continuing from Sec. 4, we provide the functional dependencies and example questions used in our binary questions and multiple-choice questions in Table 7 and Table 8, respectively. As noted in the main text, our main focus is to transform existing relational databases into questions, and the performance of an LLM on these questions may vary depending on the data fidelity of a given database.

Table 7: FDs and binary question examples of the 5 datasets.

| Dataset | FD | Example Question |
|---|---|---|
| *Movie* | *director, star, released year* → *movie title* | *Is there a movie, released in ⟨2009⟩, starring ⟨CCH Pounder⟩ where ⟨James Cameron⟩ is the director?* |
| *Soccer* | *club, jersey number, nationality* → *player name (2019 year only)* | *Is there a soccer player from ⟨Portugal⟩ who played for ⟨Juventus⟩ with uniform number ⟨7⟩ in ⟨Juventus⟩ in 2019?* |
| *Airport* | *latitude, longitude* → *airport name* | *Is there an airport located at latitude ⟨-34.7833⟩ and longitude ⟨-72.0508⟩?* |
| *Music* | *music title, released year* → *artist name* | *Is there an artist or group who sang a song titled ⟨"Day After Day"⟩ in ⟨1971⟩?* |
| *Book* | *author, published date* → *book title* | *Is there a book written by ⟨Adams, Stacy Hawkins⟩ that was published in ⟨October, 2004⟩?* |

---

[*]All datasets and codes are available at: https://github.com/DILAB-KAIST/ERBench.

Table 8: FDs and multiple-choice question examples of the 5 datasets. We generate three versions of questions with rephrased choices to measure the average performance of LLMs. The term "false", "inaccurate", or "wrong" is used when asking for the incorrect option, which is preceded by "Q: ". Each attribute of an entity is described in a short phrase that follows "Option $n$: ", where $n$ represents the respective option number.

| Dataset | FD | Example Question |
|---|---|---|
| Movie | *movie title, released year*
→ *director, country of origin, genre (animation, non-animation)* | *Q: What is the false option about the movie ⟨Avatar⟩ released in year ⟨2009⟩? Provide an explanation.*
*Option 1: It was directed by ⟨James Cameron⟩.*
*Option 2: It was produced in the country ⟨USA⟩.*
*Option 3: It is an ⟨animation⟩ movie.*

*A:* |
| Soccer | *player name (2019 year only)*
→ *club, jersey number, nationality, league* | *Q: What is the false option about soccer player named ⟨E. Hazard⟩? Provide an explanation.*
*Option 1: He played for ⟨Real Madrid CF⟩ in 2019.*
*Option 2: His uniform number was ⟨10⟩ in 2019.*
*Option 3: He was born in ⟨Belgium⟩.*
*Option 4: He played in ⟨Spain Primera Division⟩ during the year 2019.*

*A:* |
| Airport | *airport name*
→ *ICAO shortcode, latitude, longitude, country code* | *Q: What's the inaccurate option about the airport ⟨Torca⟩? Provide an explanation.*
*Option 1: ICAO Shortcode of the airport is ⟨SCLI⟩.*
*Option 2: Latitude of the airport is ⟨-34.7833⟩.*
*Option 3: Longitude of the airport is ⟨-72.0508⟩.*
*Option 4: Country code of the airport is ⟨US⟩.*

*A:* |
| Music | *music title, artist name*
→ *released year, genre (pop/rock, hip hop/raggae, country/folk, blues/jazz)* | *Q: What is the wrong option regarding the song ⟨"Day After Day"⟩ of the artist ⟨Badfinger⟩? Provide an explanation.*
*Option 1: The released year of the song is ⟨1971⟩.*
*Option 2: The song is categorized as ⟨country/folk⟩ genre.*

*A:* |
| Book | *book title, author*
→ *published month, published year, publisher name* | *Q: What's the false option about the book titled ⟨Speak to My Heart⟩ written by an author named ⟨Adams, Stacy Hawkins⟩? Provide an explanation.*
*Option 1: Published month of the book is ⟨October⟩.*
*Option 2: Published year of the book is ⟨1979⟩.*
*Option 3: Published by the publisher named ⟨Revell⟩.*

*A:* |

# B   Appendix – More Details on Experiments

## B.1   Case Study with different hallucination metrics

Continuing from Sec. 4, we add a case study (Tables 9–12) that demonstrates how each of the hallucination metrics **A** (answer accuracy), **R** (rationale accuracy), **AR** (answer & rationale accuracy), and **H** (hallucination rate) are useful.

Table 9: Case study where **A** is correct, and **R** is incorrect. Here, the model's answer is correct as "Yes", but the airport at the corresponding latitude and longitude is "Catsa Airport", not "Nosara Airport", hence the model's rationale is incorrect. We would like to distinguish the answer and rationale performances by using the **A** and **R** metrics, respectively.

---

**CASE STUDY FOR GPT-4, AIRPORT DATASET**

**Q:** Is there an airport located at latitude 10.5169439316 and longitude -85.5655593872?
**A:** Yes.
**R:** This location corresponds to the Nosara Airport in Costa Rica.

---

Table 10: Case study where **A** is incorrect, and **R** is correct. Here, the model's answer is incorrect, but its rationale contains the right keyword "Titanic". Again, we would like to distinguish the answer and rationale performances by using the **A** and **R** metrics, respectively.

---

**CASE STUDY FOR GPT-3.5, MOVIE DATASET**

**Q:** Is it true that there are no movies, released in 1997, starring Leonardo DiCaprio where James Cameron is the director?
**A:** Yes.
**R:** The only movie that DiCaprio and Cameron worked on together was "Titanic", which was released in 1997, but DiCaprio was the actor and Cameron was the director.

---

Table 11: Case study where **AR** is needed. Here, the model's answer and rationale are both correct. We would like to be more conservative and only measure the cases when both the answer and rationale are correct. The **AR** metric is used for this purpose. The gold rationale is in bold text.

---

**CASE STUDY FOR CLAUDE-3-SONNET, MUSIC DATASET**

**Q:** Is there an artist or group who sang a song titled that's the way love goes in 1993?
**A:** Yes
**R:** The song was recorded by American singer **Janet Jackson** and released as the lead single from her fifth studio album, "janet."...

---

Table 12: Case study where **H** is needed. Here, the answer is not given by the model. If an LLM is not sure about its answer, it may be better to not give one at all [11]. The **H** metric is used to measure the proportion of incorrect outputs while ignoring such non-answers.

---

**CASE STUDY FOR GEMINI-PRO, BOOK DATASET**

**Q:** Is there a book written by Francis, Dick and Francis, Felix that was published in September, 2007?
**A:** Unsure
**R:** There is no information available about a book written by Francis, Dick and Francis, Felix that was published in September 2007. It is possible that such a book exists, but without further research, it is impossible to say for sure.

---

## B.2 Assessing Internal Knowledge of LLMs

Continuing from Sec. 4, we provide more details on our approach to assess an LLM's internal knowledge. We directly ask the LLM if it knows about an entity and its attributes in a given relation. Here, the goal is not to assess if the LLMs have a deep, philosophical understanding of the entity, but rather to evaluate their mechanical abilities to represent and describe the entity and its attributes. The input prompts for these questions to assess known entities are different for binary and multiple-choice questions, as we explain below. For simplicity, let us assume that we have a *Movie* relation with an FD *title*, *year* → *director*, *length*.

**Binary Questions.** For the binary questions, we use one input prompt per entity that concatenates the following questions:

- One question that asks the left-hand side (LHS) attributes of the FD
- Multiple questions that ask the right-hand side (RHS) attributes of the FD

For example, for the *Movie* relation, we use one input prompt *"Do you know about the movie ⟨title⟩ released in ⟨year⟩? If yes, is the movie directed by ⟨director⟩? If yes, does the movie have the length ⟨length⟩?"*. Due to the concatenation, we can check whether the LLM knows all attribute values of

an entity in the given FD via a single API call. We additionally use a system prompt *"Answer the following question in yes or no. Be concise"* to get *Yes* or *No* answers from the LLM. An entity is considered to be known by the LLM if and only if it answers with only *Yes* to this input prompt.

**Multiple-choice Questions.** For the multiple-choice questions, we use multiple input prompts per entity that ask the following questions:

- One question that asks the LHS attributes of the FD
- Multiple questions that ask the RHS attributes of the FD along with LHS attributes

For example, for the *Movie* relation, we use 3 input prompts: (1) *"Do you know about the movie ⟨title⟩ released in ⟨year⟩?"*; (2) *"Is the movie ⟨title⟩ released in ⟨year⟩ directed by ⟨director⟩"*; and (3) *"Does the movie ⟨title⟩ released in ⟨year⟩ have the length ⟨length⟩?"*. The reason we do not use one concatenated input prompt as in the binary questions is that LLMs tend to give trivial *No* answers as the number of attributes increases. Since multiple-choice questions naturally utilize multiple attributes to offer various options (see example questions in Table. 8), we opt to use several succinct input prompts instead of a single, concatenated one. We observe this approach helps to prevent trivial *No* responses and enhances the scalability of LLM's internal knowledge assessment as the number of options increases. The system prompt and the answer evaluation are identical to those of binary questions.

## B.3 Number of Known Entities

Continuing from Sec. 4, we provide the number of entities known by each LLM for various tasks. We show the number of known entities per LLM when evaluating single-hop questions (Table 13), single-hop questions with fine-tuning (Table 14), multi-modal questions (Table 15), and multi-hop questions (Table 16).

Table 13: Number of entities of the 5 datasets known by each LLM using the binary (BN) and multiple-choice (MC) single-hop questions. If the LLM knows too few entities (less than 20), we exclude it when computing the number of commonly known entities (# Common).

| Model | Movie | | Soccer | | Airport | | Music | | Book | |
|---|---|---|---|---|---|---|---|---|---|---|
| | BN | MC | BN | MC | BN | MC | BN | MC | BN | MC |
| GPT-3.5 | 1,352 | 1,338 | 1,007 | 721 | 1,293 | 844 | 1,068 | 571 | 1,211 | 989 |
| GPT-4 | 1,266 | 1,358 | 1,280 | 662 | 636 | 91 | 625 | 327 | 636 | 60 |
| Llama2 | 390 | 642 | 128 | 1 | 1,011 | 1 | 238 | 43 | 650 | 0 |
| Gemini-Pro | 1,020 | 1,203 | 859 | 636 | 1,208 | 472 | 1,359 | 602 | 1,485 | 846 |
| Claude-3-Sonnet | 656 | 960 | 653 | 15 | 574 | 40 | 145 | 69 | 195 | 8 |
| Mistral | 426 | 1,284 | 459 | 840 | 1,496 | 1,390 | 1,186 | 678 | 855 | 649 |
| # Total | 1,485 | 1,485 | 1,500 | 1,500 | 1,500 | 1,500 | 1,500 | 1,500 | 1,500 | 1,500 |
| # Common | 191 | 557 | 67 | 210 | 309 | 19 | 54 | 16 | 34 | 20 |

Table 14: Number of entities of the 5 datasets known by GPT-3.5 after fine-tuning with (1) only the *Soccer* dataset and (2) the combined dataset of *Movie*, *Soccer*, *Music*, and *Book*.

| Model | Movie | Soccer | Airport | Music | Book |
|---|---|---|---|---|---|
| GPT-3.5 | 1,352 | 1,007 | 1,293 | 1,068 | 1,211 |
| GPT-3.5 + FT w/ Soccer | 1,006 | 1,308 | 930 | 313 | 670 |
| GPT-3.5 + FT w/ 4 Datasets | 438 | 314 | 971 | 137 | 109 |
| # Total | 1,485 | 1,500 | 1,500 | 1,500 | 1,500 |

Table 15: Number of entities of 2 datasets known by GPT-4V and Gemini-Pro-Vision using the binary (BN) and multiple-choice (MC) multimodal questions.

| Model | Movie | | Soccer | |
| --- | --- | --- | --- | --- |
| | BN | MC | BN | MC |
| GPT-4V | 1,330 | 1,380 | 1,178 | 641 |
| Gemini-Pro-V | 1,235 | 1,143 | 1,408 | 577 |
| # Total | 1,485 | 1,485 | 1,500 | 1,500 |

Table 16: Number of entities of 2 datasets known by each LLM using the multi-hop questions.

| Model | Movie & Director | Soccer & Olympic |
| --- | --- | --- |
| GPT-3.5 | 1,192 | 1,470 |
| GPT-4 | 1,050 | 1,453 |
| Llama2 | 435 | 1,164 |
| Gemini-Pro | 1,201 | 1,471 |
| Claude-3-Sonnet | 818 | 845 |
| Mistral | 2 | 198 |
| # Total | 1,485 | 1,500 |

## B.4 More Single-hop Question Results

Continuing from Sec. 4.1, we perform the other two types of evaluations. First, all the LLMs are evaluated with questions that all the LLMs have knowledge of in Table 17. As a result, most of the LLMs show improved performances, which is likely due to the exclusion of long-tail entities that are typically challenging to answer or reason correctly. Second, all the LLMs are evaluated with all questions, regardless of their knowledge in Table 18. As a result, the LLMs show a notable degradation in performance. The results of the other two types of evaluations are consistent with the observations highlighted in the main text: (1) rationale accuracy is lower than answer accuracy, and (2) the performances of LLMs varies depending on the question type. These results again underscore the importance of benchmarks that focus more on the LLMs' rationale and provide diverse question types.

Table 17: LLM performances using the binary basic (BN$_{(Y)}$), binary negated (BN$_{(N)}$), and multiple-choice (MC) questions on the 5 datasets. For each LLM, we evaluate with the questions whose entities are commonly known by all the LLMs. "n/a" means the LLM knows too few entities (less than 20) for the result to be meaningful; see Sec. B.3 for the number of entities known by each LLM. Lower **H** values are better, whereas higher **A**, **R**, and **AR** values are better.

| Model | Metric | Movie | | | Soccer | | | Airport | | | Music | | | Book | | |
| --- | --- | --- | --- | --- | --- | --- | --- | --- | --- | --- | --- | --- | --- | --- | --- | --- |
| | | BN$_{(Y)}$ | BN$_{(N)}$ | MC | BN$_{(Y)}$ | BN$_{(N)}$ | MC | BN$_{(Y)}$ | BN$_{(N)}$ | MC | BN$_{(Y)}$ | BN$_{(N)}$ | MC | BN$_{(Y)}$ | BN$_{(N)}$ | MC |
| GPT-3.5 | **A** | **.97** | .20 | **.99** | .40 | .00 | .86 | .17 | .00 | .88 | .85 | .54 | **1.0** | **.94** | .03 | **.73** |
| | **R** | .95 | .31 | .98 | .45 | .07 | .69 | .04 | .00 | .82 | .78 | .59 | **.96** | **.42** | .03 | **.28** |
| | **AR** | **.95** | .17 | .98 | .37 | .00 | .63 | .04 | .00 | .82 | .76 | .50 | **.96** | **.42** | .03 | .22 |
| | **H** (↓) | **.03** | .80 | **.01** | .60 | 1.0 | .14 | .71 | .96 | .12 | .11 | .46 | .00 | .03 | .94 | .27 |
| GPT-4 | **A** | .92 | .90 | **.99** | **.70** | .54 | .92 | .71 | .13 | **.93** | **.94** | .85 | **1.0** | .70 | .09 | .48 |
| | **R** | **.96** | **.96** | .99 | **.96** | **.88** | .75 | **.29** | **.23** | **.91** | **.83** | **.78** | **.96** | **.42** | .03 | **.28** |
| | **AR** | .91 | .87 | **.99** | **.69** | .51 | **.75** | **.26** | **.05** | **.91** | **.83** | **.76** | **.96** | .39 | .03 | **.28** |
| | **H** (↓) | .08 | .10 | **.01** | .27 | .03 | **.02** | .29 | .81 | **.07** | .06 | **.00** | **.00** | .06 | **.00** | .02 |
| Llama2 | **A** | .10 | **1.0** | .93 | .03 | **1.0** | n/a | .00 | **1.0** | n/a | .93 | **1.0** | .94 | .06 | **1.0** | n/a |
| | **R** | .64 | **.96** | .97 | .22 | .67 | n/a | .00 | .05 | n/a | .61 | .54 | .92 | .15 | **.29** | n/a |
| | **AR** | .10 | **.96** | .92 | .03 | **.67** | n/a | .00 | **.05** | n/a | .59 | .54 | .88 | .03 | **.29** | n/a |
| | **H** (↓) | .90 | **.00** | .07 | .97 | **.00** | n/a | 1.0 | **.00** | n/a | .07 | **.00** | .06 | .94 | **.00** | n/a |
| Gemini-Pro | **A** | .63 | .00 | .95 | .03 | .00 | **.94** | .00 | .00 | .84 | .85 | .20 | **1.0** | .09 | .00 | .62 |
| | **R** | .87 | .43 | .99 | .16 | .16 | .55 | .00 | .00 | .39 | .67 | .33 | .88 | .09 | .03 | .20 |
| | **AR** | .61 | .00 | .95 | .01 | .00 | .53 | .00 | .00 | .39 | .63 | .17 | .88 | .09 | .00 | .20 |
| | **H** (↓) | .30 | 1.0 | .05 | **.22** | .39 | .06 | **.00** | .38 | .16 | **.00** | .10 | **.00** | **.00** | .03 | .38 |
| Claude | **A** | .56 | .95 | .99 | .45 | .04 | n/a | .56 | .01 | .83 | .59 | .46 | .88 | .48 | .00 | n/a |
| | **R** | .95 | .94 | **1.0** | .64 | .06 | n/a | .12 | .11 | .75 | .51 | .38 | .92 | .36 | .09 | n/a |
| | **AR** | .54 | .32 | **.99** | .43 | .04 | n/a | .07 | .01 | .67 | .41 | .38 | .81 | .33 | .00 | n/a |
| | **H** (↓) | .44 | .66 | **.01** | .27 | **.00** | n/a | **.00** | **.00** | .17 | .23 | **.00** | .12 | **.00** | **.00** | n/a |
| Mistral | **A** | .61 | 1.0 | .78 | .46 | 1.0 | .45 | **.84** | 1.0 | .47 | .74 | 1.0 | .50 | .24 | .97 | .43 |
| | **R** | .59 | .73 | .80 | .33 | .48 | .25 | .00 | .00 | .02 | .20 | .22 | .56 | .06 | .06 | .07 |
| | **AR** | .42 | .73 | .71 | .19 | .48 | .21 | .00 | .00 | .02 | .20 | .22 | .19 | .41 | .03 | .06 |
| | **H** (↓) | .39 | **.00** | .22 | .54 | **.00** | .55 | .16 | **.00** | .53 | .22 | **.00** | .50 | .73 | .03 | .57 |

Table 18: LLM performances using the binary basic (BN(Y)), binary negated (BN(N)), and multiple-choice (MC) questions on the 5 datasets. For each LLM, we evaluate with all questions regardless of the LLM knowledge of entities. Lower **H** values are better, whereas higher **A**, **R**, and **AR** values are better.

| Model | Metric | Movie BN(Y) | Movie BN(N) | Movie MC | Soccer BN(Y) | Soccer BN(N) | Soccer MC | Airport BN(Y) | Airport BN(N) | Airport MC | Music BN(Y) | Music BN(N) | Music MC | Book BN(Y) | Book BN(N) | Book MC |
|---|---|---|---|---|---|---|---|---|---|---|---|---|---|---|---|---|
| GPT-3.5 | A | **.83** | .06 | .94 | .35 | .00 | .74 | .12 | .00 | .64 | .47 | .16 | **.82** | **.71** | .00 | **.52** |
| | R | **.78** | .10 | .94 | .25 | .01 | .51 | .01 | .00 | .35 | .29 | .14 | .48 | .11 | .01 | .12 |
| | AR | **.77** | .05 | .93 | .21 | .00 | .45 | .01 | .00 | .35 | .26 | .11 | .44 | .10 | .00 | .10 |
| | H (↓) | **.17** | .94 | .06 | .62 | 1.0 | .26 | .69 | .97 | .36 | .37 | .81 | .18 | .07 | .96 | .48 |
| GPT-4 | A | .58 | .45 | **.96** | .45 | .18 | **.81** | .61 | .09 | **.76** | **.58** | .38 | .76 | .33 | .01 | .28 |
| | R | .76 | .69 | .95 | **.64** | **.46** | .58 | **.11** | **.08** | **.44** | **.46** | **.31** | .48 | **.14** | .01 | .15 |
| | AR | .57 | .44 | **.95** | .38 | .16 | **.57** | **.09** | **.01** | **.44** | **.36** | **.28** | **.47** | **.12** | .01 | **.14** |
| | H (↓) | .42 | .53 | **.04** | .38 | .03 | **.06** | .39 | .82 | **.07** | .37 | .03 | **.11** | .07 | .01 | **.06** |
| Llama2 | A | .02 | **1.0** | .85 | .01 | **1.0** | .63 | .00 | **1.0** | .38 | .55 | **1.0** | .73 | .02 | **1.0** | .44 |
| | R | .31 | **.82** | .92 | .07 | .36 | .41 | .00 | .01 | .22 | .13 | .16 | .43 | .02 | **.06** | .09 |
| | AR | .02 | **.82** | .81 | .00 | **.36** | .32 | .00 | **.01** | .21 | .12 | .16 | .40 | .00 | **.06** | .07 |
| | H (↓) | .98 | **.00** | .15 | .99 | **.00** | .37 | 1.0 | **.00** | .62 | .45 | **.00** | .27 | .98 | **.00** | .56 |
| Gemini-Pro | A | .21 | .00 | .88 | .00 | .00 | .80 | .00 | .00 | .59 | .51 | .02 | .80 | .00 | .00 | .45 |
| | R | .57 | .24 | .94 | .06 | .04 | .50 | .00 | .00 | .24 | .13 | .05 | .40 | .00 | .00 | .10 |
| | AR | .20 | .00 | .87 | .00 | .00 | .45 | .00 | .00 | .24 | .11 | .01 | .38 | .00 | .00 | .10 |
| | H (↓) | .43 | 1.0 | .12 | **.20** | .32 | .20 | **.00** | .43 | .41 | **.02** | .10 | .20 | **.02** | .02 | .55 |
| Claude | A | .15 | .07 | .95 | .16 | .23 | .74 | .50 | .02 | .69 | .22 | .19 | **.82** | .13 | .00 | .38 |
| | R | .75 | .74 | **.97** | .23 | .01 | **.60** | .04 | .03 | .35 | .11 | .06 | **.50** | .07 | .02 | **.18** |
| | AR | .15 | .07 | .94 | .11 | .01 | .52 | .02 | .00 | .34 | .06 | .05 | .46 | .05 | .00 | .13 |
| | H (↓) | .85 | .89 | .05 | .24 | **.00** | .22 | .02 | **.00** | .31 | .50 | .01 | .17 | .15 | .01 | .33 |
| Mistral | A | .29 | **1.0** | .69 | **.50** | .99 | .40 | **.71** | **1.0** | .12 | .41 | .96 | .56 | .13 | .99 | .38 |
| | R | .28 | .39 | .71 | .06 | .13 | .20 | .00 | .00 | .09 | .03 | .03 | .30 | .00 | .01 | .04 |
| | AR | .13 | .39 | .59 | .04 | .13 | .17 | .00 | .00 | .05 | .02 | .03 | .11 | .00 | .01 | .02 |
| | H (↓) | .71 | **.00** | .31 | .50 | .01 | .60 | .29 | **.00** | .87 | .55 | **.00** | .43 | .86 | .01 | .62 |

## B.5 More Analyses on Rationale Verification Accuracy

Continuing from Sec. 4.2, where we provide results comparing correctness of ERBench to human rationale evaluation, we provide results comparing the correctness of ERBench's verification to GPT-Judge, where we use GPT-4 [1] for the GPT-Judge [7] with the prompts shown in Sec. C.3. We first check that GPT-Judge makes similar evaluations as humans by making a comparison of 3,000 random responses. The similarity turns out to be 93.8%, which we view as reliable. We then apply GPT-Judge on 15,000 question and answer pairs, for each model. The results are shown in Table 19. ERBench achieves correctness higher than 94% on average and higher than 88% for any model, which again supports our claim that ERBench's strategy of looking for critical keywords is effective.

Table 19: ERBench's verification accuracy compared to GPT-Judge.

| | GPT-3.5 | GPT-4 | Llama2 | Gemini | Claude-3 | Mistral |
|---|---|---|---|---|---|---|
| Acc (%) | 96.6 | 91.5 | 96.6 | 96.2 | 88.4 | 97.7 |

We also perform an error analysis on what ERBench misses and identify three categories of corner cases: (1) the rationale has incorrect information about an inferred entity that is not related to the question; (2) there is an entity resolution error within the rationale; and (3) there is a logical-self contradiction in the rationale. Most corner cases fall under (3), and a reasonable solution is to employ general self-consistency methods [65, 66] on top of ERBench. The main idea is to let the LLM give multiple different responses to a question and then choose the most consistent one. We currently configure the LLM to give a deterministic single response to each question following the convention of other hallucination benchmarks [7, 11, 36], but this can easily be extended if needed. The corresponding results are out of our scope, hence we excluded the results.

## B.6 More Extended Multiple-Choice Question Results

Continuing from Sec. 4.1, we provide results of extending the multiple-choice questions with the *None of the above* option. Fig. 2 shows the resulting performance of GPT-3.5 and GPT-4 on the 3 datasets. Both GPT-3.5 and GPT-4 show decreased answer accuracies as the proportion of *None of the above* options increases. Notably, GPT-4 demonstrates a more stable performance compared to GPT-3.5 where the answer accuracy decreases by no more than 4%.

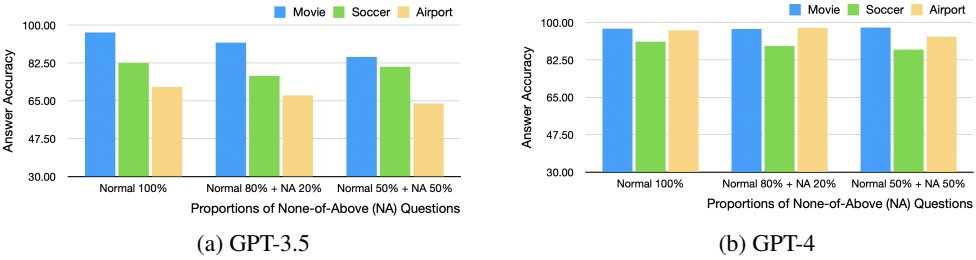

(a) GPT-3.5                    (b) GPT-4

Figure 2: GPT-3.5 and GPT-4 answer accuracy across domains with varying portions of the *None of the above* option for multiple-choice questions.

## B.7 Multi-hop Question Construction

Continuing from Sec. 4.3, we provide more details on the multi-hop question construction. For multi-hop analysis, we construct 2-hop and 3-hop questions by joining relations using foreign key constraints (FKCs) as outlined in Sec. 3.3. For 2-hop questions, we join two relations – *Movie* and *Director* – using an FKC on the director's name (i.e., *director* in *Movie* and *director name* in *Director*). For 3-hop questions, we join three relations – *Soccer*, *Club*, *Olympic* – using the two FKCs: (1) club name to join *Soccer* and *Club* (i.e., *club* in *Soccer* and *club name* in *Club*) and (2) city name to join *Club* and *Olympic* (i.e., *located city* in *Club* and *city name* in *Olympic*; see more dataset details in Sec. A.1). Given the joined relations, we ask questions to infer foreign key values along with the targeted answer – see example questions in Table 20.

Table 20: FDs, FKs, and binary multi-hop question examples of the 2 datasets.

| Dataset | FD | Example Question |
|---------|----|------------------|
| *Movie & Director* | Movie: *movie title → director* (**FK**) 
 Director: *director name → birth year* | *Was the director who directed the movie titled ⟨Avatar⟩ that was released in ⟨2009⟩ born in the ⟨1950s⟩?* |
| *Soccer & Olympic* | Soccer: *player name → club* (**FK**) *(in 2019)* 
 Club: *club name → located city* (**FK**) 
 Olympic: *city name → hosted year* | *Did the city, where the soccer club, ⟨L. Messi⟩ played for in 2019, is located in, hosted the Summer Olympics?* |

## B.8 More Analyses on Multi-Hop Question Results

Continuing from Sec. 4.3, we further highlight ERBench's advantages by providing more analyses on the multi-hop dataset in Sec. 4.3. We introduce two metrics:

- $\mathbf{Pr}(r_{i+1}|r_i)$: the conditional probability of the rationale at hop $i + 1$ being correct given that the rationale at hop $i$ is correct
- $\mathbf{Pr}(r_{i+1}|\neg r_i)$: the conditional probability of rationale at hop $i + 1$ being correct given that the rationale at hop $i$ is incorrect

The results are shown in Table 21. $\mathbf{Pr}(r_{i+1}|r_i)$ is significantly higher than $\mathbf{Pr}(r_{i+1}|\neg r_i)$ for all cases, which means that hop $i$'s correctness largely influences hop $i + 1$'s correctness. Thus it is important for initial hops to be correct to avoid any snowballing of incorrectness.

We observe that for Gemini on the Movie & Director dataset, the correctness of a prior hop has relatively less influence on the correctness of subsequent hops. We provide a representative case

Table 21: LLM performances for additional analyses using binary basic and negated multi-hop question w/wo CoT prompting on 2 datasets – see Sec. B.8 for details. All the numbers for the *Movie & Director* dataset (2-hop) denote $\Pr(r_2|r_1)$. For the *Soccer & Olympic* dataset (3-hop), the numbers on the left are the values for $\Pr(r_2|r_1)$, while those on the right are $\Pr(r_3|r_2)$. "n/a" means the denominator of the metric is too small for the result to be meaningful (less than 4). For each question type, we mark the best performance in bold among all models w/wo CoT prompting.

| Model | Metric | Movie & Director | | | | Soccer & Olympic | | | |
| | | w/o CoT | | w/ CoT | | w/o CoT | | w/ CoT | |
| | | BN(Y) | BN(N) | BN(Y) | BN(N) | BN(Y) | BN(N) | BN(Y) | BN(N) |
|---|---|---|---|---|---|---|---|---|---|
| GPT-3.5 | $\Pr(r_{i+1}|r_i)$ | .95 | **.96** | .93 | .95 | **1.0**/.57 | .95/.79 | **1.0/1.0** | **1.0**/.94 |
| | $\Pr(r_{i+1}|\neg r_i)$ | .04 | .03 | **.06** | .00 | .10/.00 | .15/.00 | .33/.04 | .31/.02 |
| GPT-4 | $\Pr(r_{i+1}|r_i)$ | .96 | **.96** | **.97** | .95 | .99/.96 | .99/.97 | **1.0**/.98 | **1.0**/.97 |
| | $\Pr(r_{i+1}|\neg r_i)$ | n/a | n/a | n/a | n/a | .35/.00 | .27/.00 | **.52**/.00 | **.55**/.00 |
| Llama2 | $\Pr(r_{i+1}|r_i)$ | .92 | .93 | **.97** | .92 | **1.0**/.85 | **1.0**/.81 | **1.0**/.95 | **1.0**/.93 |
| | $\Pr(r_{i+1}|\neg r_i)$ | .00 | .00 | .00 | .00 | .35/.00 | .18/.00 | .24/.02 | .20/**.04** |
| Gemini-Pro | $\Pr(r_{i+1}|r_i)$ | .71 | .61 | .61 | .34 | **1.0**/n/a | .96/.90 | **1.0**/.98 | **1.0**/.73 |
| | $\Pr(r_{i+1}|\neg r_i)$ | .01 | .03 | .00 | .00 | .00/.00 | .01/.00 | .28/**.07** | .11/.00 |
| Claude-3-Sonnet | $\Pr(r_{i+1}|r_i)$ | .90 | .92 | .93 | .92 | .99/.88 | .97/.94 | **1.0**/.98 | **1.0**/.98 |
| | $\Pr(r_{i+1}|\neg r_i)$ | n/a | **.22** | n/a | **.22** | .22/.00 | .03/.00 | .05/.00 | .02/.00 |

study in Table 22, where Gemini correctly retrieves the director's name (hop 1) of the corresponding movie, but fails to retrieve his birth year (hop 2) and thus fails to reason across multi-hops.

Table 22: Case study for which the model fails to retrieve correct information for the subsequent hop, even when it successfully retrieves relevant information for the prior hop. **Q:** is the prompt that we input to the model, **R:** is the model's rationale, and **A:** is the model's answer. Gemini knows that the director of the corresponding movie is Rob Marshall, but fails to retrieve his birth year. One interesting point is that Gemini knows the birth year of the director, when asked separately.

---

**CASE STUDY FOR GEMINI, MOVIE & DIRECTOR DATASET**

**Q:** Was the director who directed the movie titled Pirates of the Caribbean: On Stranger Tides that was released in 2011 born in the 1960s?
**R:** The director of Pirates of the Pirates of the Caribbean: On Stranger Tides is Rob Marshall. There is no publicly available information about his date of birth, so it is not possible to say for sure whether he was born in the 1960s.

*(However, Gemini does know the birth year of Rob Marshall if we ask separately.)*

**Q:** Which year was the director Rob Marshall born in?
**A:** 1960.

---

## B.9   More Few-Shot Prompting Results

Continuing from Sec. 4.5, we provide more details of the few-shot results. When evaluating Mistral, we reduce the size of each dataset to 150 entities due to its extensive inference time. For the binary questions, the numbers of entities known by Mistral in the *Movie*, *Soccer*, *Airport*, *Music*, and *Book* datasets are 72, 81, 150, 123, and 86, respectively. For the multiple-choice questions, the numbers are 140, 99, 140, 62, and 62, respectively. Table 23 shows the LLM performances in the few-shot setting on the 5 datasets. Compared to the single-hop results in Table 2, the demonstrations in few-shot setting significantly improve the performances of most LLMs, especially for the binary negated prompts. Rather than comparing performances among LLMs, we highlight the overall improvements, as the relative performances among LLMs can greatly vary depending on the quantities and qualities of the given demonstrations. For clarity, we also present all the demonstration prompts of the 5 datasets used in our experiments in Sec. C.2.

Table 23: LLM performances in few-shot setting on the 5 datasets. The questions and the LLMs used for the evaluation are identical to those in Table 2. The size of the 5 datasets was reduced for Mistral's evaluation due to its extensive inference time. "n/a" means the LLM knows too few entities (less than 20) for the result to be meaningful; see Sec. B.3 for the number of entities known by each LLM. Lower **H** values are better, whereas higher **A**, **R**, and **AR** values are better.

| Model | Metric | Movie BN(Y) | BN(N) | MC | Soccer BN(Y) | BN(N) | MC | Airport BN(Y) | BN(N) | MC | Music BN(Y) | BN(N) | MC | Book BN(Y) | BN(N) | MC |
|---|---|---|---|---|---|---|---|---|---|---|---|---|---|---|---|---|
| GPT-3.5 | A | **.79** | .70 | .95 | .81 | .63 | .86 | .21 | .20 | .71 | .80 | .77 | .93 | .91 | .76 | .57 |
| | R | .87 | .69 | .96 | .72 | .60 | .70 | .04 | .03 | .44 | .47 | .43 | .66 | .22 | .20 | .22 |
| | AR | **.75** | .66 | .95 | .60 | .48 | .68 | .02 | .02 | .44 | .45 | .42 | .66 | .21 | **.18** | .21 |
| | H (↓) | **.21** | .30 | .05 | .18 | .35 | .14 | .79 | .78 | .29 | .18 | .22 | .07 | .03 | .22 | .43 |
| GPT-4 | A | .62 | .65 | .98 | .68 | .59 | **.94** | .84 | .91 | **.97** | **.82** | .89 | **.98** | .61 | .40 | **.89** |
| | R | **.90** | **.82** | .98 | **.82** | **.80** | **.86** | **.24** | **.24** | **.91** | .78 | **.76** | .87 | .35 | **.31** | **.57** |
| | AR | .61 | .64 | .98 | .62 | **.51** | **.86** | **.24** | **.24** | **.91** | **.72** | **.72** | **.87** | .27 | .17 | **.57** |
| | H (↓) | .38 | .35 | .02 | .32 | .41 | **.06** | .16 | .09 | **.03** | **.17** | .11 | **.02** | .30 | .56 | **.11** |
| Llama2 | A | .67 | .74 | .96 | **.88** | .62 | n/a | **1.0** | .50 | n/a | .81 | .57 | .94 | **.95** | .24 | n/a |
| | R | .75 | .69 | .95 | .59 | .43 | n/a | .01 | .00 | n/a | **.81** | .57 | **.94** | **.95** | .24 | n/a |
| | AR | .61 | .66 | .94 | .54 | .37 | n/a | .01 | .00 | n/a | .34 | .24 | .71 | .09 | .04 | n/a |
| | H (↓) | .33 | .26 | .04 | .12 | .38 | n/a | **.00** | .50 | n/a | .19 | .43 | .06 | .04 | .76 | n/a |
| Gemini-Pro | A | .20 | .04 | .96 | .74 | .24 | .91 | .40 | .00 | .72 | .74 | .59 | .88 | .88 | .20 | .55 |
| | R | .53 | .62 | .95 | .56 | .41 | .68 | .01 | .00 | .35 | .17 | .16 | .54 | .11 | .09 | .13 |
| | AR | .19 | .03 | .95 | .48 | .13 | .67 | .01 | .00 | .35 | .15 | .11 | .54 | .11 | .05 | .13 |
| | H (↓) | .59 | .96 | .04 | **.10** | .67 | .09 | **.00** | .46 | .28 | .19 | .38 | .12 | **.01** | .72 | .45 |
| Claude-3 -Sonnet | A | .47 | .41 | **.99** | .79 | .58 | n/a | .85 | .69 | .92 | .61 | .47 | .91 | .66 | .28 | n/a |
| | R | .80 | .72 | **.99** | .74 | .65 | n/a | .08 | .07 | .79 | .37 | .29 | .88 | .25 | .14 | n/a |
| | AR | .46 | .40 | **.99** | **.63** | .46 | n/a | .08 | .06 | .78 | .37 | .28 | .81 | .24 | .14 | n/a |
| | H (↓) | .52 | .53 | **.01** | .20 | .18 | n/a | .15 | .14 | .08 | .21 | .10 | .08 | .16 | .04 | n/a |
| Mistral | A | .43 | **1.0** | .34 | .32 | **1.0** | .58 | .73 | **1.0** | .27 | .39 | **.99** | .49 | .14 | **.99** | .36 |
| | R | .74 | .81 | .39 | .21 | .35 | .33 | .01 | .00 | .03 | .02 | .01 | .34 | .00 | .00 | .01 |
| | AR | .39 | **.81** | .31 | .07 | .35 | .29 | .01 | .00 | .00 | .01 | .01 | .29 | .00 | .00 | .01 |
| | H (↓) | .57 | **.00** | .66 | .68 | **.00** | .42 | .27 | **.00** | .73 | .59 | **.00** | .51 | .84 | **.01** | .64 |

## B.10 Knowledge Augmentation Results

Continuing from Sec. 4.5, we provide more results on knowledge augmentation. We implement a simple knowledge augmentation method using the LangChain API [31], which adds a short summary of an entity or attribute's Wikipedia page to its corresponding single-hop questions used in Sec. 4.1. We also inject the following system prompt to avoid excessive reliance on external texts: *The first few passages are hints, that may not contain all relevant information. Answer the following question with your own knowledge getting help from the first few passages if possible.*

Table 24 shows the resulting performances of GPT-3.5 and GPT-4 on the 2 datasets. Compared to the single-hop results, the performances degrade for the binary questions, but improve for the multiple-choice questions. We note that knowledge augmentation can be more challenging in the binary questions compared to the multiple-choice questions, as an entity is explicitly mentioned in the multiple-choice questions (e.g., *What is the false option of entity e?*), but not in the binary questions (e.g., *Is there an entity which has the attribute values of x and y?*). These results show how ERBench can properly stress test knowledge augmentation approaches, which may perform well on one question type, but not necessarily on others.

## C  Appendix – More Related Work

Continuing from Sec. 5, we provide additional related work.

**LLM Hallucination Detection Methods**   We summarize LLM hallucination detection methods that focus more on improving the LLMs instead of evaluating them. First, there are methods [67–69] that use intrinsic uncertainty metrics like token probability and entropy to detect hallucinations. However, access to token-level probability distributions of LLM responses may be difficult, especially if only a limited external API is provided. In comparison, ERBench does not rely on such metrics. Next, there are methods [70, 71] that retrieve relevant information from external databases to provide

Table 24: GPT-3.5 and GPT-4 performances using the binary basic (BN(Y)), binary negated (BN(N)), and multiple-choice (MC) questions with knowledge augmentation (KA) on 2 datasets. See Sec. B.3 for the number of entities known by each LLM. Lower **H** values are better, whereas higher **A**, **R**, and **AR** values are better. For each question type, we mark the best performance in bold among all models w/wo KA.

| Model | Metric | Movie | | | | | | Soccer | | | | | |
| | | w/o KA | | | w/ KA | | | w/o KA | | | w/ KA | | |
| | | BN(Y) | BN(N) | MC | BN(Y) | BN(N) | MC | BN(Y) | BN(N) | MC | BN(Y) | BN(N) | MC |
|---|---|---|---|---|---|---|---|---|---|---|---|---|---|
| GPT-3.5 | **A** | **.85** | .06 | .97 | .57 | .34 | **.99** | .35 | .00 | .83 | .17 | .00 | .89 |
| | **R** | **.81** | .11 | .96 | .51 | .35 | .63 | .28 | .02 | .60 | .04 | .00 | .37 |
| | **AR** | **.80** | .05 | .96 | .50 | .22 | .62 | .24 | .00 | .55 | .04 | .00 | .35 |
| | **H** (↓) | **.15** | .94 | .03 | .37 | .44 | **.01** | .64 | 1.0 | .17 | .10 | .52 | .11 |
| GPT-4 | **A** | .65 | .51 | .97 | .57 | **.56** | **.99** | **.47** | **.19** | .91 | .01 | .03 | **.96** |
| | **R** | **.81** | **.76** | .97 | .68 | .64 | **.98** | **.70** | **.49** | **.69** | 02 | .02 | .57 |
| | **AR** | .64 | .50 | **.97** | .56 | **.54** | **.97** | **.41** | **.17** | **.69** | .01 | .02 | .57 |
| | **H** (↓) | .35 | .47 | .03 | .39 | **.34** | **.01** | .38 | .04 | **.02** | **.02** | **.00** | .04 |

evidence when assessing the factuality of a given LLM response. However, these additional evidence retrieval steps can be erroneous [72] and may not be sufficient for the assessment. ERBench on the other hand takes a different approach where it builds upon an existing relational database where we already have knowledge on how to evaluate the responses. Finally, there are methods [73, 74] that use LLMs to self-check themselves, for example by prompting them to evaluate their previous responses. However, the detection performance of these methods can be highly dependent on the given LLM's performance. In comparison, ERBench's evaluation depends less on the LLM's performance.

**LLM Benchmarks from Knowledge Graphs** Numerous LLM benchmarks use knowledge graphs (KGs) [11, 75, 76] for generating evaluation samples, but ERBench use relational databases (RDBs), which have fundamental differences with KGs. While both RDBs and KGs can store large amounts of data and enable scalable benchmarks by converting data into factual questions, they rely on different data models. RDBs are based on the relational data model and assume a fixed schema, which enables strong data integrity based on database design theory; KGs are based on the graph data model and have a schema-less design, which means the format is more flexible, but it may be more challenging to maintain the data integrity.

The key idea of ERBench is to utilize data integrity constraints in RDBs for more effective LLM hallucination evaluation. In particular, ERBench uses (1) functional dependencies (FDs) to automatically pinpoint critical keywords and assess LLM reasoning, and (2) foreign key constraints (FKCs) to systematically construct complex multi-hop questions. These benefits are not easily supported by KGs, as integrity constraints are unique to RDBs. For example, without strong signals like FDs, it can be nontrivial to determine which keywords to focus on in the rationale, leading to most KG-based methods to verify only final answers [11, 77, 78]. Moreover, without FKCs, constructing arbitrarily long multi-hop questions involving multiple connected entities becomes challenging, and existing KG-based methods [41, 79] often require manual curation of bridge entities [41] or logical rules [79] to make the connections. ERBench thus complements KG-based methods, offering unique benefits and enabling new types of analysis through RDBs.

## C.1 Multimodal Prompts

Continuing from Sec. 4.4, we provide the multimodal questions used in our experiments. Table 25 and Table 26 show the FDs and the questions of two datasets for the binary and multiple-choice questions, respectively. Due to the restriction policies of vision models, particularly those prohibiting the use of facial images, we make slight modifications to the question templates for the *Movie* dataset. Similar to Sec. 4.1, we also negate the binary questions to evaluate both *Yes* and *No* answers of the LLMs.

Table 25: FDs and binary questions for multimodal tasks.

| Dataset | FD | Question |
|---|---|---|
| *Movie* | *director, star, released year*
→ *movie title, movie poster (image)* | *Is the movie, released in ⟨year⟩, starring ⟨star⟩ where ⟨dir⟩ is the director the same movie as the movie with the movie poster as the given image?* |
| *Soccer* | *nationality, jersey number, club logo (image)*
→ *player name (2019 year only)* | *Is there a soccer player from ⟨country⟩ who played for the club in the image with uniform number ⟨no⟩ in the club in the image in 2019?* |

Table 26: FDs and multiple-choice questions for multimodal tasks.

| Dataset | FD | Question |
|---|---|---|
| *Movie* | *movie poster (image)*
→ *director, country of origin, genre (animation, non-animation)* | *Q: What's the inaccurate option about the movie with the movie poster as the given image? Provide an explanation.*
*Option 1: Directed by ⟨dir⟩*
*Option 2: Produced in the country ⟨country⟩*
*Option 3: Has the genre of ⟨genre⟩ movie.*

*A:* |
| *Soccer* | *player name (2019 year only)*
→ *club logo (image), jersey number, nationality, league* | *Q: What's the inaccurate option about soccer player? Provide an explanation.*
*Option 1: Played for the club in the image in 2019.*
*Option 2: Wore jersey number ⟨no⟩ in 2019.*
*Option 3: Born in ⟨country⟩.*
*Option 4: Participated in league named ⟨league⟩ during the year 2019.*

*A:* |

## C.2 Demonstration Prompts

Continuing from Sec. 4.5, we provide all demonstration prompts used in the few-shot and Chain-of-Thought [13] (CoT) analyses. Tables 27–36 show the demonstration prompts in the few-shot setting, and Tables 37–38 show the demonstration prompts with CoT. We manually design all demonstration prompts to have balanced distributions w.r.t. the answer type (e.g., *Yes/No* and *Option* numbers) to reduce any potential bias in the resulting LLM performances to certain answers.

Table 27: Demonstration prompts for the *Movie* dataset used in few-shot prompting before asking each binary question. To generate questions that lead to a negative/positive response for the existence/negation prompt respectively, the *released year* of the movie is modified.

---

**PROMPTS FOR MOVIE DATASET**

**Q:** Is there a movie, released in 1975, starring Al Pacino where Sidney Lumet is the director?
**A:** Yes, the movie, among various movies Al Pacino starred in 1975, Sideny Lumet directed the movie "Dog Day Afternoon".

**Q:** Is it true that there are no movies, released in 2005, starring Ben Affleck where Kevin Smith is the director?
**A:** Yes, it is true. However, there is a movie,"Chasing Amy" that Kevin Smith directed, where Ben Affleck starred in, which was released in 1997.

**Q:** Is it true that there are no movies, released in 2019, starring Kangho Song where Bong Joon-ho is the director?
**A:** No, it is not true. The movie "Parasite", which was released in 2019 was directed by Bong Joon-ho, where Kangho Song starred in the movie.

**Q:** Is there a movie, released in 1997, starring Jason Biggs where Paul Weitz is the director?
**A:** No, however there is a movie, "American Pie" that Paul Weitz directed, where Jason Biggs starred in, which was released in 1999.

**Q:** Is there a movie, released in 1979, starring George Lazenby where Peter R. Hunt is the director?
**A:** No, however there is movie, "On Her Majesty's Secret Service" that George Lazenby starred in, where Peter R. Hunt is the director, which was released in 1969.

**Q:** Is it true that there are no movies, released in 2005, starring Leonardo DiCaprio where Martin Scorsese is the director?
**A:** Yes, it is true. However, there is a movie, "Gangs of New York" that Martin Scorsese directed and Leonardo DiCaprio starred in, which was released in 2002.

**Q:** Is it true that there are no movies, released in 1952, starring Robert Taylor where Richard Thorpe is the director?
**A:** No, it is not true. The movie, "Ivanhoe" that Richard Thorpe directed, where Robert Taylor appeared in the movie was released in 1952.

**Q:** Is there a movie, released in 1982, starring Dustin Hoffman where Sydney Pollack is the director?
**A:** Yes, the movie, among various movies that Sydney Pollack directed in 1982, Dustin Hoffman starred in the movie, "Tootsie".

---

Table 28: Demonstration prompts for the *Soccer* dataset used in few-shot prompting before asking each binary question. To generate questions that lead to a negative/positive response for the existence/negation prompt respectively, the *nationality* of the soccer player is modified.

---

**PROMPTS FOR SOCCER DATASET**

**Q:** Is there a soccer player from Republic of Korea who played for Valencia CF with uniform number 16 in Valencia CF in 2019?
**A:** Yes, Kangin Lee, a South Korean footballer, was a member of Valencia CF and wore the uniform number 16 while playing for Valencia CF in 2019.

**Q:** Is it true that there are no soccer players from Italy who played for Everton with uniform number 27 in Everton in 2019?
**A:** No, it is not true. There is an Italian soccer player, M. Kean, who was a member of Everton and wore the uniform number 27 while playing for Everton in 2019.

**Q:** Is there a soccer player from Brazil who played for Vissel Kobe with uniform number 10 in Vissel Kobe in 2019?
**A:** No. Łukasz Józef Podolski (L. Podolski), who was a member of Vissel Kobe and wore the uniform number 10 while playing for Vissel Kobe in 2019 is from Germany, not Brazil.

**Q:** Is it true that there are no soccer players from Italy who played for Crystal Palace with uniform number 13 in Crystal Palace in 2019?
**A:** Yes, it is true. The soccer player, who was a member of Crystal Palace wearing the uniform number 13 while playing for Everton in 2019 was W. Hennessey. He is from Wales, not Italy.

**Q:** Is there a soccer player from Portugal who played for Paris Saint-Germain with uniform number 26 in Paris Saint-Germain in 2019?
**A:** No. Jesé Rodríguez Ruiz (Jesé), a Spanish soccer player, was a member of Paris Saint-Germain and wore the uniform number 26 while playing for Paris Saint-Germain in 2019. He is from Spain, not Portugal.

**Q:** Is there a soccer player from Italy who played for Torino F.C. with uniform number 29 in Torino F.C. in 2019?
**A:** Yes, Lorenzo De Silvestri (L. De Silvestri), an Italian footballer, was a member of Torino F.C. and wore the uniform number 29 while playing for Torino F.C. in 2019.

**Q:** Is it true that there are no soccer players from England who played for Burnley with uniform number 28 in Burnley in 2019?
**A:** No, it is not true. There is a English footballer, M. Lowton, who was a member of Burnley and wore the uniform number 2 while playing for Burnley in 2019.

**Q:** Is it true that there are no soccer players from Japan who played for Philadelphia Union with uniform number 10 in Philadelphia Union in 2019?
**A:** Yes it is true. The soccer player, who was a memebr of Philadelphia Union while wearing the uniform number 10 when playing for Philadelphia Union in 2019 was Marco Jhonfai Fabián De La Mora (M. Fabián).

---

Table 29: Demonstration prompts for the *Airport* dataset used in few-shot prompting before asking each binary question. To generate questions that lead to a negative/positive response for the existence/negation prompt respectively, *latitude* and *longitude* corresponding to places other than airports are added.

---

**PROMPTS FOR AIRPORT DATASET**

**Q:** Is there an airport located at latitude 33.7756 and longitude -84.3963?
**A:** No, there are no airports located at latitude 33.7756 and longitude -84.3963. This location corresponds to Georgia Institute of Technology, located in Atlanta, Georiga, United States of America.

**Q:** Is it true that there are no airports located at latitude -33.4445 and longitude -70.6510?
**A:** Yes, it is true. There are no airports located at latitude -33.4445 and longitude -70.6510. This location corresponds to University of Chile, located in Santiago, Región Metropolitana, Chile.

**Q:** Is it true that there are no airports located at latitude 48.1180 and longitude 16.5663?
**A:** No, it is not true. There is an airport located at latitude 48.1180 and longitude 16.5663. The airport is called Vienna International Airport and is located in Vienna, Austria.

**Q:** Is there an airport located at latitude -32.2189 and longitude 148.5697?
**A:** Yes, there is an airport located at latitude -32.2189 and longitude 148.5697. The airport is called Dubbo Regional Airport, located in Dubbo, New South Wales, Australia.

**Q:** Is there an airport located at latitude 25.420738 and longitude 51.490154?
**A:** No, there are no airports located at latitude 25.420738 and longitude 51.490154. This location corresponds to Lusail Stadium, located in Lusail, Qatar.

**Q:** Is it true that there are no airports located at latitude -32.9277 and longitude 18.4237?
**A:** Yes, it is true. There are no airports located at latitude -32.9277 and longitude 18.4237. This location corresponds to District Six Museum, located in District Six, Cape Town, South Africa.

**Q:** Is there an airport located at latitude 37.469101 and longitude 126.450996?
**A:** Yes, there is an airport located at latitude 37.469101 and longitude 126.450996. The airport is called Incheon International Airport and is located in Incheon, Republic of Korea.

**Q:** Is it true that there are no airports located at latitude 49.1951 and longitude -123.1788?
**A:** No, it is not true. There is an airport located at latitude 49.1951 and longitude -123.1788. The airport is called Vancouver International Airport, located in Richmond, British Columbia, Canada.

---

Table 30: Demonstration prompts for the *Music* dataset used in few-shot prompting before asking each binary question. To generate questions that lead to a negative/positive response for the existence/negation prompt respectively, the *released year* of the music is modified.

---

**PROMPTS FOR MUSIC DATASET**

**Q:** Is there an artist or group who sang a song titled solitude standing in 1985?
**A:** No, there is no such artist or group. However, there is an artist who sang a song titled "Solitude Standing" in 1987. The song is by the artist Suzanne Vega.

**Q:** Is it true that no artists nor groups sang a song titled i'm sitting on top of the world in 1953?
**A:** No, it is not true. Les Paul sang a song titled "I'm Sitting On Top Of The World" in 1953.

**Q:** Is it true that no artists nor groups sang a song titled you can't hurry love in 1976?
**A:** Yes, it is true. However, there is an artist who sang a song titled "You Can't Hurry Love" in 1966. The song is by the group The Supremes.

**Q:** Is there an artist or group who sang a song titled rebel girl in 1992?
**A:** Yes, there is an artist who sang a song titled "Rebel Girl" in 1992. The song is by the group Bikini Kill.

**Q:** Is it true that no artists nor groups sang a song titled dissident aggressor in 1972?
**A:** Yes, it is true. However, there is an artist who sang a song titled "Dissident Aggressor" in 1977. The song is by the artist Judas Priest.

**Q:** Is there an artist or group who sang a song titled young black male in 1988?
**A:** No, there is no such artist or group. However, there is an artist who sang a song titled "Young Black Male" in 1991. The song is by the artist 2Pac.

**Q:** Is it true that no artists nor groups sang a song titled because you live in 2004?
**A:** No, it is not true. Jesse Mccartney sang a song titled "Because You Live" in 2004.

**Q:** Is there an artist or group who sang a song titled the age of worry in 2012?
**A:** Yes, there is an artist who sang a song titled "The Age Of Worry" in 2012. The song is by the artist John Mayor.

---

Table 31: Demonstration prompts for the *Book* dataset used in few-shot prompting before asking each binary question. To generate questions that lead to a negative/positive response for the existence/negation prompt respectively, the *published year* of the book is modified.

---

**PROMPTS FOR BOOK DATASET**

**Q:** Is there a book written by Hodge, Paul that was published in October, 1984?
**A:** Yes, there is a book written by Paul Hodge that was published in October, 1984. The book is titled "The Universe of Galaxies".

**Q:** Is it true that there are no books written by Mosel, Ted that were published in January, 1978?
**A:** No, it is not true. There is a book written by Ted Mosel that was published in January, 1978. The book is titled "Leading Lady: The World and Theatre of Katharine Cornell".

**Q:** Is it true that there are no books written by Blanchard, Kenneth H. that were published in June, 1998?
**A:** Yes, it is true. There are no books written by Kenneth H. Blanchard that were published in June, 1998. There is a book, written by Kenneth H. Blanchard that was published in a similar date, June, 1999. The book is titled "The Heart of a Leader".

**Q:** Is there a book written by Terzian, James P. that was published in February, 1959?
**A:** No, there are no books written by James P. Terzian that was published in February, 1959. There is a book, written by James P. Terzian that was published in a similar date, January, 1959. The book is titled "Caravan from Ararat".

**Q:** Is there a book written by Bowman, Martin that was published in November, 1994?
**A:** No, there are no books written by Martin Bowman that was published in November, 1994. There is a book, written by Martin Bowman that was published in a similar date, November, 1993. The book is titled "Spirits in the Sky".

**Q:** Is it true that there are no books written by Wylie, Philip that were published in March, 1965?
**A:** Yes, it is true. There are no books written by Philip Wylie that were published in March, 1965. There is a book , written by Philip Wylie that was published in a similar date, January, 1965. The book is titled "They both were naked".

**Q:** Is it true that there are no books written by Stallwood, Kim W. that were published in December, 2019?
**A:** No, it is not true. There is a book written by Kim W. Stallwood that was published in December, 2019. The book is titled "Speaking Out for Animals: True Stories About People Who Rescue Animals".

**Q:** Is there a book written by Suskind, Ron that was published in January, 2004?
**A:** Yes, there is a book written by Ron Suskind that was published in January, 2004. The book is titled "The Price of Loyalty: George W. Bush, the White House, and the Education of Paul O'Neill".

---

Table 32: Demonstration prompts for the *Movie* dataset used in few-shot prompting before asking each multiple-choice question.

---

**PROMPTS ғᴏʀ MOVIE DATASET**

**Q:** What's the inaccurate option about the movie Step Up Revolution released in year 2012?
Provide an explanation.
Option 1: Directed by David Wain.
Option 2: Produced in the country USA.
Option 3: Has the genre of non-animation movie.

**A:** Option 1: Directed by David Wain.
The movie "Step Up Revolution" released in year 2012 was directed by Scott Speer.

**Q:** What is the false option about the movie The Watcher released in year 2000?
Provide an explanation.
Option 1: It was directed by Joe Charbanic.
Option 2: It was produced in the country China.
Option 3: It is a non-animation movie.

**A:** Option 2: It was produced in the country China.
The movie "The Watcher" released in year 2000 was produced in the country USA.

**Q:** What is the wrong option regarding the movie Snakes on a Plane released in year 2006?
Provide an explanation.
Option 1: The name of the Director is David R. Ellis.
Option 2: The movie was produced in the country Germany.
Option 3: The movie is an animation movie.

**A:** Option 3: The movie is an animation movie.
The movie "Snakes on a Plane" released in year 2006 is a non-animation movie.

---

Table 33: Demonstration prompts for the *Soccer* dataset used in few-shot prompting before asking each multiple-choice question.

---

**PROMPTS ꜰᴏʀ SOCCER DATASET**

**Q:** What's the inaccurate option about soccer player K. Dolberg? Provide an explanation.
Option 1: Played for Real Valladolid CF in 2019.
Option 2: Wore jersey number 25 in 2019.
Option 3: Born in Denmark.
Option 4: Participated in leauge named Holland Eredivisie during the year 2019.

A: Option 1: Played for Real Valladolid CF in 2019.
Kasper Dolberg, commonly known as K. Dolberg, played in AFC Ajax in 2019.

**Q:** What is the false option about soccer player named Palhinha? Provide an explanation.
Option 1: He played for SC Braga in 2019.
Option 2: His uniform number was 10 in 2019.
Option 3: He was born in Portugal.
Option 4: He played in Portuguese Liga Zon Sagres during the year 2019.

**A:** Option 2: His uniform number was 10 in 2019.
The uniform number of João Palhinha was 60 during the year 2019.

**Q:** What is the wrong option regarding the soccer player A. Barboza? Provide an explanation.
Option 1: He participated in Club Atlético Independiente during the year 2019.
Option 2: His jersey number during 2019 was 26.
Option 3: His birthplace is Italy.
Option 4: He participated in Argentina Primera División during the year 2019.

**A:** Option 3: His birthplace is Italy.
Alexander Barboza, commonly known as A. Barboza, was born in Argentina.

**Q:** What's the inaccurate option about soccer player T. Abraham? Provide an explanation.
Option 1: Played for Chelsea in 2019.
Option 2: Wore jersey number 9 in 2019.
Option 3: Born in England.
Option 4: Participated in leauge named French Ligue 2 during the year 2019.

**A:** Option 4: Participated in leauge named French Ligue 2 during the year 2019.
Tammy Abraham, commonly known as T. Abraham, participated in English Premier League during 2019.

Table 34: Demonstration prompts for the *Airport* dataset used in few-shot prompting before asking each multiple-choice question.

---

**PROMPTS ғᴏʀ AIRPORT DATASET**

**Q:** What's the inaccurate option about the airport Arlanda? Provide an explanation.
Option 1: ICAO Shortcode of the airport is ESSB.
Option 2: Latitude of the airport is 59.649.
Option 3: Longitude of the airport is 17.923.
Option 4: Country code of the airport is SE.

**A:** Option 1: ICAO Shortcode of the airport is ESSB.
The ICAO code for Arlanda airport is ESSA.

**Q:** What is the false option about the airport Tshane? Provide an explanation.
Option 1: The abbreviated form (ICAO) for the airport is FBTE.
Option 2: The latitude of the airport is 37.017.
Option 3: The longitude of the airport is 21.882.
Option 4: The country code of the airport is BW.

**A:** Option 2: The latitude of the airport is 37.017.
The latitude of Tshane airport is approximately -24.017.

**Q:** What is the wrong option regarding the airport Oum el Bouaghi? Provide an explanation.
Option 1: The ICAO shortcode for the airport is the same with DABO.
Option 2: The airport is located at 35.879 latitude.
Option 3: The airport is located at -110.200 longitude.
Option 4: The airport has a country code of DZ.

**A:** Option 3: The airport is located at -110.200 longitude.
The longitude of Oum el Bouaghi airport is approximately 7.270.

**Q:** What's the inaccurate option about the airport Bruny Island? Provide an explanation.
Option 1: ICAO Shortcode of the airport is YBYI.
Option 2: Latitude of the airport is -43.234.
Option 3: Longitude of the airport is 147.380.
Option 4: Country code of the airport is US.

A: Option 4: Country code of the airport is US.
Bruny Island airport is located in Australia, not the United States. Therefore, its country code is AU.

---

Table 35: Demonstration prompts for the *Music* dataset used in few-shot prompting before asking each multiple-choice question.

---

**PROMPTS ғᴏʀ MUSIC DATASET**

**Q:** What's the inaccurate option about the song goodbye june of the artist get happy?
Provide an explanation.
Option 1: The song was released in 2008.
Option 2: The genre of the song is blues/jazz.

**A:** Option 1: The song was released in 2008.
The song "Get Happy" by the artist Goodbye June was released in 2018.

**Q:** What is the false option about the song it's a feeling of the artist toto?
Provide an explanation.
Option 1: The song was released in the year 1982.
Option 2: The song belongs to country/folk genre.

**A:** Option 2: The song belongs to country/folk genre.
The song "It's a feeling" by the artist Toto belongs to pop/rock genre.

---

Table 36: Demonstration prompts for the *Book* dataset used in few-shot prompting before asking each multiple-choice question.

---

**PROMPTS FOR BOOK DATASET**

**Q:** What's the inaccurate option about the book titled The Case of the Ancient Astronauts, written by an author named Gallagher, I.J? Provide an explanation.
Option 1: Published month of the book is February.
Option 2: Published year of the book is 1977.
Option 3: Published by the publisher named Heinemann/Raintree.

**A:** Option 1: Published month of the book is February.
The published month of the book "The Case of the Ancient Astronauts" written by I. J. Gallagher is January.

**Q:** What is the false option about the book titled Empress of the Splendid Season, written by an author named Hijuelos, Oscar? Provide an explanation.
Option 1: The book was published in the month January.
Option 2: The book was published in the year 2008.
Option 3: The book was published by the publisher named Harper Flamingo.

**A:** Option 2: The book was published in the year 2008.
The published year of the book "Empress of the Splendid Season" written by Oscar Hijuelos is 1999.

**Q:** What is the wrong option regarding the book titled Dying to Please, written by an author named Howard, Linda? Provide an explanation.
Option 1: The published month of the book is April.
Option 2: The published year of the book is 2002.
Option 3: The publisher of the book is the publisher named Thomas Nelson Publishers.

**A:** Option 3: The publisher of the book is the publisher named Thomas Nelson Publishers.
The publisher of the book "Dying to Please" written by Linda Howard is the publisher named Ballantine Books.

---

Table 37: Demonstration prompts with Chain-of-Thought for *Soccer w/ Olympic* Dataset. Players are manually sampled from the dataset to generate questions that lead to a negative/positive response for the existence/negation prompt, respectively.

---

**PROMPTS FOR SOCCER w/ Olympic DATASET**

**Q:** Did the city, where the soccer club, Kangin Lee played for in 2019, is located in, hosted the Summer Olympics?
**A:** No, Kangin Lee played for the soccer club Valencia CF in 2019. Valencia CF is located in the city, Valencia in Spain. Valencia has never hosted the Summer Olympics. The answer is no.

**Q:** Did the city, where the soccer club, M. Kean played for in 2019, is located in, never hosted the Summer Olympics?
**A:** Yes, M. Kean played for the soccer club Everton in 2019. Everton is located in the city, Liverpool in England. Liverpool has never hosted the Summer Olympics. The answer is yes.

**Q:** Did the city, where the soccer club, W. Hennessey played for in 2019, is located in, never hosted the Summer Olympics?
**A:** No, W. Hennessey played for the soccer club Crystal Palace in 2019. Crystal Palace is located in the city, London in England. London has hosted the Summer Olympics in 1908, 1948, and 2012. The answer is no.

**Q:** Did the city, where the soccer club, Jesé played for in 2019, is located in, hosted the Summer Olympics?
**A:** Yes, Jesé played for the soccer club Paris Saint-Germain in 2019. Paris Saint-Germain is located in the city, Paris in Frace. Paris has hosted the Summer Olympics in 1900 and 1924. The answer is yes.

**Q:** Did the city, where the soccer club, J. Kluivert played for in 2019, is located in, hosted the Summer Olympics?
**A:** Yes, J. Kluivert played for the soccer club Roma in 2019. Roma is located in the city, Rome in Italy. Rome has hostedthe Summer Olympics in 1960. The answer is yes.

**Q:** Did the city, where the soccer club, Diego Oliveira played for in 2019, is located in, never hosted the Summer Olympics?
A: No, Diego Oliveira played for the soccer club FC Tokyo in 2019. FC Tokyo is located in the city, Tokyo in Japan. Tokyo has hosted the Summer Olympics in 1964 and 2020. The answer is no.

**Q:** Did the city, where the soccer club, L. Podolski played for in 2019, is located in, hosted the Summer Olympics?
**A:** No, L. Podolski played for the soccer club Vissel Kobe in 2019. Vissel Kobe is located in the city, Kobe in Japan. Kobe has never hosted the Summer Olympics. The answer is no.

**Q:** Did the city, where the soccer club, M. Fabián played for in 2019, is located in, never hosted the Summer Olympics?
**A:** Yes, M. Fabián played for the soccer club Philadelphia Union in 2019. Philadelphia Union is located in the city, Philadelphia in the United States. Philadelphia has never hosted the Summer Olympics. The answer is yes.

Table 38: Demonstration prompts with Chain-of-Thought for *Movie w/ Director* Dataset. To generate questions that lead to a negative/positive response for the existence/negation prompt, the *decade of born year* of the director is modified.

---

**PROMPTS FOR MOVIE w/ DIRECTOR DATASET**

**Q:** Was the director who directed the movie titled Dog Day Afternoon that was released in 1975 born in the 1920s?
**A:** Yes, the director who directed the movie titled Dog Day Afternoon that was released in 1975 was born in the 1920s. Sidney Lumet, the director of Dog Day Afternoon, was born on June 25, 1924. The answer is yes.

**Q:** Is it true that the director who directed the movie titled Chasing Amy that was released in 1997 was not born in the 1960s?
**A:** Yes, it is true. Kevin Smith, who directed the movie titled Chasing Amy that was released in 1997 was born in the 1970s, not in the 1960s. He was born on August 2, 1970. The answer is yes.

**Q:** Is it true that the director who directed the movie titled Parasite that was released in 2019 was not born in the 1960s?
**A:** No, it is not true. Bong Joon-ho, who directed the movie titled Parasite that was released in 2019 was born in the 1960s. He was born on September 14, 1969. The answer is no.

**Q:** Was the director who directed the movie titled American Pie that was released in 1999 born in the 1970s?
**A:** No, the director who directed the move titled American Pie that was released in 1999 was born in the 1960s. Paul Weitz, the director of American Pie, was born on November 19, 1965. The answer is no.

**Q:** Is it true that the director who directed the movie titled On Her Majesty's Secret Service that was released in 1969 was not born in the 1930s?
**A:** Yes, it is true. The director who directed the movie titled On Her Majesty's Secret Service that was released in 1969 was born in the 1920s, not in the 1930s. Peter R. Hunt, the director of On Her Majesty's Secret Service was born on March 11, 1925. The answer is yes.

**Q:** Was the director who directed the movie titled Ivanhoe that was released in 1952 born in the 1900s?
**A:** No, Richard Thorpe, who directed the movie titled Ivanhoe that was released in 1952 was born in the 1890s, not in the 1900s. He was born on February 24, 1896. The answer is no.

**Q:** Was the director who directed the movie titled Gangs of New York that was released in 2002 born in the 1940s?
**A:** Yes, Martin Scorsese, who directed the movie titled Gangs of New York that was released in 2002 was born in the 1940s. He was born on November 17, 1942. The answer is yes.

**Q:** Is it true that the director who directed the movie titled Tootsie that was released in 1982 was not born in the 1930s?
**A:** No, it is not true. The director who directed the movie titled Tootsie that was released in 1982 was born in the 1930s. He was born on July 1, 1934. The answer is no.

---

## C.3 GPT-Judge Prompts

Table 39: Prompt for GPT-Judge (GPT-4). The {entity}, {attributes} differ across datasets based on the properties used for the corresponding task. For instance, for the *Movie* dataset, the {entity} is *movie*, and the {attributes} are *director*, *star*, and *released year*.

---

**PROMPT FOR GPT-JUDGE**

Answer in yes or no.
**A1**: {model's outputted rationale}
**A2**: {ground truth rationale}

Are the two answers, **A1** and **A2**, referring to the same {entity} with the same properties, {attributes}? If there is no {entity} names mentioned in both **A1** and **A2**, output yes. If only one of **A1** and **A2** mention a {entity} name, output no. If both **A1** and **A2** mention {entity} name, answer only looking at **A1** and **A2** without your knowledge.

---

