# OpenReview forum: "ERBench: An Entity-Relationship based Automatically Verifiable Hallucination Benchmark for Large Language Models"
_NeurIPS.cc/2024/Datasets_and_Benchmarks_Track — NeurIPS 2024 Track Datasets and Benchmarks Spotlight_

### Official Review · Reviewer_oNmd · 2024-07-02
**Using rich relational databases to generate challenging tests of LLM reasoning**

**Rating:** 7
**Confidence:** 4
**Clarity:** The paper is well written and clear.

**Review:**

The approach of using complex relational databases to create challenging LLM queries is something that I fully endorse. While the examples provided in the paper are strong and motivating, there is a potential to explore even more subtle relationships that can explore the metalepsic relationships between historical figures and the boundaries and contingencies of fiction. For example, there are numerous instances of actors playing biographical figures from history. Consider the science fiction movie that posits that H G Wells, an author, actually created a "real" time machine. The fact that much of post-modern entertainment focuses on the artistic process, there are numerous examples of actors "playing themselves" in a fictional setting. All of this compounds the usual contingent logic of "you can kill a vampire with a stake through the heart" versus "vampires do not exist."

This paper is a good start in the direction of testing the kind of relations that people have no problem: that movie about the thing with the guy that was in that other movie etc. Some of the example queries approach this level of complexity, but the aggregate comparisons between the LLMs don't really say much about relative power the LLMs bring to their analysis. Perhaps some kind of breakdown based on the number of hops might start to show where these differences start and diverge.

**Strengths:**

Strong and extensible framework for robustly testing LLMs on useful tasks.

Comprehensive analysis of multiple open-source and commercial large language models.

Interesting results on prompt fine tuning.

**Additional Feedback:**

No additional feedback.

**Correctness:**

The authors indicate that at least some of the data is from Wikipedia crawls. The details, and the dates of such data extraction are important and should be given in more completeness.

**Documentation:**

The details about the collection and processing of the data, and the relational schema could be documented more completely.

**Ethics:**

No concerns.

**Limitations:**

The authors identify the potential for incorrect entity matches leading to lower recall rates, but there are also dangers with the underlying data being incorrect or incomplete.

**Opportunities For Improvement:**

More analysis and synthesis of the results - what are the specific types of errors and are errors made by different models on similar problems?

More details about the underlying data. It's very important that the underlying data be correct, and even data from Wikipedia or IMDB is known to contain difficult to resolve errors. (To say nothing of actors with uncredited roles or other misattributions both intentional and unintentional.)

**Relation To Prior Work:**

There is not a significant review of related work, and this could be improved.

**Summary And Contributions:**

Using a variety of resources in relation database form, questions with answers require multiple joins and multiple intermediate results are created to probe and test the limits of LLM question answering capabilities.

Drawing upon topics from entertainment, geography, and sports the authors propose a strong and extensible framework that is used to evaluate a variety of popular large language models. These tests show, in some cases, wide gaps in the performance, including unique measures of the hallucination rates.

They also probe chain of thought reasoning by checking for the mentions of intermediate results that are necessary for getting a multi-phase question answered correctly. There are also experiments where Wikipedia descriptions are included to help the LLM, but these surprisingly seem to have a deleterious effect.

Overall I think this is a fine addition to set of problems that LLMs should use during testing.

---

> ### Author Rebuttal · Authors · 2024-08-16
>
> We appreciate your thoughtful comments and useful suggestions. We are happy to address them below and in our revision.
>
> &nbsp;
> > **Comment 1.  Exploring the metalepsic relationships**
>
> We agree that extending ERBench to explore more subtle relationships such as metalepsic relationships is an interesting direction. As long as one can construct a database with such relationships, ERBench can be used as is. Otherwise, there are also extended notions of functional dependencies (FDs) like conditional functional dependencies (CFDs) [1], which are FDs that hold only under certain circumstances (e.g., FDs that only hold in a certain country) and can be tailored to new relationships.
>
> [1] Bohannon, Philip, et al. "Conditional functional dependencies for data cleaning." ICDE 2007.
>
> &nbsp;
> > **Comment 2. Performance breakdown with number of hops**
>
> We agree that a further breakdown analysis across LLMs would be useful. Multi-hop questions may indeed cause some divergence in performance. In Sec. B.7, we observed that the correctness of a prior hop usually has a significant influence on the correctness of subsequent hops. However, Gemini used on the Movie & Director dataset seems to be an outlier where the influence is much smaller as shown in Table 16 (i.e., the $\Pr (r_{i+1} | r_i)$ values are quite low). In the representative example below, Gemini correctly retrieves the director’s name of the corresponding movie, but fails to retrieve his birth year and thus fails to reason across multi-hops.
>
> ---
>
> *Example (Gemini, Movie & Director Dataset)*
>
> Question: Was the director who directed the movie titled Pirates of the Caribbean: On Stranger Tides that was released in 2011 born in the 1960s?
>
> Rationale:  The director of Pirates of the Caribbean: On Stranger Tides is Rob Marshall. There is no publicly available information about his date of birth, so it is not possible to say for sure whether he was born in the 1960s.
>
> *(However, Gemini does know the birth year of Rob Marshall if we ask separately)*
>
> Question: Which year was the director Rob Marshall born in?
>
> Answer: 1960.
>
> ---
>
> We will add these results in our revision.
>
> &nbsp;
> > **Comment 3. More analysis on error cases**
>
> Thanks for your question. As discussed in Sec. B.4, the errors mainly fall into three categories: 1) incorrect information about an inferred entity, 2) entity resolution errors, and 3) logical self-contradiction. To further address your question, we compare different model responses to the same question: Is there a movie, released in 2006, starring Hugh Jackman where Brett Ratner is the director? The correct answer is “Yes” and the model rationale should include the movie title “X-men”. The LLM responses are as follows:
>
> - Gemini, GPT-3.5, Claude: responds *correctly*
>
> - GPT-4: responds with *incorrect star* information
>
> - Mistral: responds with *incorrect director* information
>
> - LLama: responds with *incorrect star/year* information
>
> We observe that although the last three models have the same error type (i.e., incorrect information about an inferred entity), their specific pieces of incorrect information may vary. We will add this more fine-grained analysis in our revision.
>
> &nbsp;
> > **Comment 4. Underlying data correctness**
>
> We fully agree that data correctness is important. In our experiments, we performed basic cleaning techniques like filtering null values, but we are aware of the vast literature of data cleaning in databases [2, 3] as well. It is also the database owner’s responsibility to come up with the correct database design including integrity constraints as we emphasized in Sec. 2.
>
> [2] Ilyas, Ihab and Chu, Xu, “Data Cleaning”, ACM 2019.
>
> [3] Rekatsinas, Theodoros et al., “HoloClean: Holistic Data Repairs with Probabilistic Inference”, VLDB 2017.
>
> &nbsp;
> > **Comment 5. Related works**
>
> Thanks for your feedback and we will cover more recent studies as follows:
>
> - More KG-based LLM evaluation methods [4, 5]
>
> These methods mainly focus on automatically verifying model *answers* using knowledge graphs (KGs). In comparison, ERBench not only verifies model answers, but also the *rationales* by leveraging FDs of relational databases (RDBs), a feature not available in KGs.
>
> [4] Feng, Chao, Xinyu Zhang, and Zichu Fei. "Knowledge solver: Teaching llms to search for domain knowledge from knowledge graphs." ArXiv 2023.
>
> [5] Qian, Cheng, Xinran Zhao, and Sherry Tongshuang Wu. ""Merge Conflicts!" Exploring the Impacts of External Distractors to Parametric Knowledge Graphs." ArXiv 2023.
>
> &nbsp;
> - More benchmarks with multi-hop questions [6, 7]
>
> These methods require *manual curation* of bridge entities to construct multi-hop questions. In comparison, ERBench *systematically identifies* bridge entities and constructs multi-hop questions by using foreign key constraints in RDBs.
>
> [6] Ho, Xanh, et al. "Constructing a multi-hop QA dataset for comprehensive evaluation of reasoning steps." COLING 2020.
>
> [7] Trivedi, Harsh, et al. "♫ MuSiQue: Multihop Questions via Single-hop Question Composition." TACL 2022.
>
> &nbsp;
> We will add the above citations in our revision.
>
> &nbsp;
> > **Comment 6. Documentation**
>
> Thanks, we will include additional details of the data collection (Sec. 4) and detailed information of the relational schemas (Sec. A.1 and Sec. A.2) in the documentation in our revision.
>
> We constructed three relations using Wikipedia crawls: Director (director name, birth year), Club (club name, located city), and Olympic (city name, hosted years). For the Director relation, we utilized the BeautifulSoup library to crawl Wikipedia pages for each director and parsed the relevant table, identifying the birth year in the row marked by the indicator "Born”. For the Club relation, we crawled and searched Wikipedia pages that have information of each soccer club. We manually extracted the located cities. Similarly for the Olympic relation, we utilized the Summer Olympics page in Wikipedia and manually generated the relations.

---

> > ### Comment · Reviewer_oNmd · 2024-08-16
> > **Acknowledgement**
> >
> > Thank you for these responses and associated changes - I think they address my concerns and improve the paper overall.

---

### Official Review · Reviewer_FFNu · 2024-07-24
**Entity-Relationship based Auto evaluation Hallucination Benchmark for LLMs**

**Rating:** 7
**Confidence:** 3
**Correctness:** Yes
**Clarity:** Yes

**Review:**

The paper demonstrates ERBench's effectiveness in assessing LLM performance across different question types, prompting methods, and fine-tuning scenarios. ERBench addresses the limitations of existing benchmarks by providing a systematic way to generate complex, automatically verifiable questions from structured data. It offers a more comprehensive and flexible approach to LLM evaluation compared to existing benchmarks, particularly in assessing the reasoning capabilities of LLMs on complex, multi-step questions. See strengths and weaknesses below.

**Strengths:**

1. This paper is well organized and well written.
2. ERBench introduces a new method for constructing LLM benchmarks using relational databases, which is a significant departure from existing approaches that typically use knowledge bases or manually constructed datasets.
3. Automatic verification is important. By leveraging database integrity constraints, ERBench can automatically verify both the correctness of LLM answers and the validity of their rationales. This is a key advancement over benchmarks that only check final answers.
4. The experiments demonstrate ERBench's effectiveness across multiple LLMs, question types, and prompting methods, providing a thorough validation of the approach.
5. ERBench's support for continuous evaluation, multimodal questions, and various prompt engineering techniques makes it highly adaptable to evolving LLM capabilities and evaluation needs.

**Additional Feedback:**

N/A

**Documentation:**

Yes

**Ethics:**

No ethical concerns

**Limitations:**

Yes

**Opportunities For Improvement:**

1. Consider adding a case study comparing different hallucination metrics evaluation methods.
2. Consider adding a human study for further validation.
3. In addition to base LLMs, consider the hallucination evaluation method for adapted LLMs.

**Relation To Prior Work:**

Yes

**Summary And Contributions:**

This paper introduces ERBench, a new approach for constructing large language model (LLM) benchmarks using relational databases based on the entity-relationship (ER) model. ERBench utilizes existing relational databases to automatically generate complex, verifiable questions for evaluating LLMs. It leverages database integrity constraints like functional dependencies and foreign key constraints to construct questions and verify LLM responses, including both answers and rationales. ERBench can generate single-hop and multi-hop questions of arbitrary complexity by joining relations using foreign keys. The benchmark is extensible, supporting continuous evaluation as databases change, multimodal questions, and various prompt engineering techniques. Experiments were conducted using 5 public databases to evaluate several popular LLMs on single-hop, multi-hop, and multimodal questions. Results show ERBench can effectively evaluate LLMs by checking both answer correctness and rationale validity using critical keywords.

---

> ### Author Rebuttal · Authors · 2024-08-16
>
> We appreciate your thoughtful comments and useful suggestions. We are happy to address them below and in our revision.
>
> &nbsp;
>
> > **Comment 1. Case study with different hallucination metrics**
>
> We add a new case study that demonstrates how each of the hallucination evaluation metrics – A (answer accuracy), R (rationale accuracy), AR (answer & rationale accuracy), and H (hallucination rate) – are useful. We will add the below results in our revision.
>
>
> *Case 1. answer is correct, but rationale is incorrect*
>
> ---
> Example (GPT-4, Airport Dataset)
>
> Question: Is there an airport located at latitude 10.5169439316 and longitude -85.5655593872?
>
> Answer: Yes.
>
> Rationale:  This location corresponds to the Nosara Airport in Costa Rica.
>
> ---
>
> Here the answer Yes is correct, but the airport at the above latitude and longitude is Catsa Airport, not Nosara Airport. We would like to distinguish the answer and rationale performances by using the A and R metrics, respectively.
>
> &nbsp;
>
> *Case 2. answer is incorrect, but rationale is correct*
>
> ---
> Example (GPT-3.5, Movie Dataset)
>
>
> Question: Is it true that there are no movies, released in 1997, starring Leonardo DiCaprio where James Cameron is the director?
>
> Answer: Yes.
>
> Rationale: The only movie that DiCaprio and Cameron worked on together was "Titanic," which was released in 1997, but DiCaprio was the actor and Cameron was the director.
>
> ---
> In this case, the answer is incorrect, but the rationale contains the right keyword “Titanic”. Again, we would like to distinguish the answer and rationale performances by using the A and R metrics, respectively.
>
> &nbsp;
>
> *Case 3. answer and rationale are both correct*
>
> ---
> Example (Claude, Music Dataset)
>
> Question: Is there an artist or group who sang a song titled that's the way love goes in 1993?
>
> Answer: Yes
>
> Rationale:  The song was recorded by American singer Janet Jackson and released as the lead single from her fifth studio album, "janet."....
>
> ---
> We would also like to be more conservative and only measure the cases when both the answer and rationale are correct as above. The AR metric is used for this purpose.
>
> &nbsp;
>
> *Case 4. answer is not given*
>
> ---
> Example (Gemini, Book Dataset)
>
> Question:  Is there a book written by Francis, Dick and Francis, Felix that was published in September, 2007?
>
> Answer: Unsure
>
> Rationale: There is no information available about a book written by Francis, Dick and Francis, Felix that was published in September 2007. It is possible that such a book exists, but without further research, it is impossible to say for sure.
>
> ---
> If an LLM is not sure about its answer, it may be better to not give one at all [1] . The H metric is used to measure the proportion of incorrect outputs while ignoring such non-answers.
>
> &nbsp;
>
> [1] Sun, Kai, et al. "Head-to-tail: How knowledgeable are large language models (llm)? AKA will llms replace knowledge graphs?." NAACL 2024.
>
> &nbsp;
> > **Comment 2. Human study**
>
> Thanks for the suggestion, and we would like to clarify that we already conducted a human study to validate the correctness of ERBench and analyze corner cases (Sec 4.2). Let us know if the reviewer expects some new study.
>
>
> &nbsp;
> > **Comment 3. Hallucination evaluation for adapted LLMs**
>
> Thanks, we did perform hallucination evaluation for fine-tuned LLMs in our paper and would like to summarize the results. First, fine-tuning GPT-3.5 on the Soccer dataset only resulted in a substantial improvement on the Soccer dataset (+0.56%p for the AR metric on average), whereas the improvements on other datasets were smaller (+0.22%p AR on average). Second, fine-tuning GPT-3.5 on four datasets – Movie, Soccer, Music, and Book – together had mixed results (Table 19). In particular, the performance on the Music dataset was much lower than when fine-tuning on the Soccer dataset only (-24.5%p AR on average), and the overall performance was not higher either. Let us know if the reviewer expects other experiments as well.

---

### Official Review · Reviewer_dgij · 2024-07-24
**A way of curating textual dataset from databases but lacks of depth.**

**Rating:** 5
**Confidence:** 3
**Correctness:** See above.
**Clarity:** See above.

**Review:**

## Quality

This paper itself is written in great detail. The result is comprehensive. But the analysis lacks depth.

## Clarity
This paper, combined with the code repository, conveys clear technical details for readers to understand and reproduce the work.

## Originality
The originality is somehow questionable. There are a number of works suggesting using knowledge bases, in particular, knowledge graphs, to benchmark the hallucination of LLMs. Given that the knowledge graph is also a relational structure, I cannot give enough credit to automatic question population from a given relational knowledge base.

## Strengths
- The evaluation provides surprisingly good details and of knowledge from five domains.
- Some part of the automatic evaluation results is also supported by human evaluation.

## Weakness
- For benchmarking hallucination, the main purpose of ERBench, the knowledge coverage is limited. One could found that each dataset is about 1,000 to 2,000. It is not clear whether these knowledge forms a challenge of LLMs in terms of hallucination.
- Some concepts are not well justified. For example, the entity known by the LLMs is ill defined. One might argue that the LLM models are  well supported.
- Despite the detailed evaluation of many levels, it is hard for readers to identify new findings derived from the benchmark apart from general descriptions of results. We have already known that stronger models will have better performance, and harder questions (multi-hop) will be more challenging.

**Strengths:**

See strength above.

**Additional Feedback:**

See above.

**Documentation:**

Not very complete but the code repository is easy to understand.
- the data and samples are well presented
- the code for generating such data from an **arbitrary** relational db is not given, which is also claimed as one major contribution of this work.

**Ethics:**

N.A

**Opportunities For Improvement:**

1. The reviewer would like to see why we need this benchmark, and how can this benchmark derives new findings / what are those findings / how can this benchmark lead to better approach for mitigate hallucination.

**Relation To Prior Work:**

See above.

**Summary And Contributions:**

Summary:

This paper introduces ER-bench, where the data from each table and joined tables are linearized into claims, stated as Functional Dependency (FD) and Foreign Key Constraints (FKC), respectively. The FD is related to single hop questions and the FKC is related to multihop questions. The authors also tested five different LLMs on such linearized data. A very comprehensive findings are reported.

Contributions:

1. This paper showcases how to general data from a relational database from two aspects, FD and FKC.
2. Comprehensive evaluations are found in five example databases.

---

> ### Author Rebuttal · Authors · 2024-08-16
>
> To Reviewer dgij (Response 2/2),
>
> > **Comment 5. Codes for arbitrary dataset**
>
> Thanks for your feedback. The code for using an arbitrary relational database is in run_qa.py. We provided detailed guidelines on how to use our framework in our source comments and documentation.  For clarification, let us assume that a user has a dataset with the functional dependency A,B,C -> D. To generate a new LLM benchmark with ERBench, the user only needs to create a question template in the form that resembles the questions in Table 6 (and Sec. B.1 for validation) and add them to the get_prompt function (validation_step function for validation) in run_qa.py. With these simple steps, one can easily generate an LLM benchmark after executing the code in order as mentioned in the code documentation. We will add these points in our revision.

---

> ### Author Rebuttal · Authors · 2024-08-17
>
> To Reviewer dgij (Response 1/2),
>
> We appreciate your insightful comments. We are happy to address them below and in our revision.
>
> &nbsp;
> > **Comment 1. Originality compared to KG**
>
> We respect your viewpoint and clarify that 1) relational databases (RDBs) are fundamentally different than knowledge graphs (KGs) because they use the relational data model and 2) ERBench’s novelty is to exploit database integrity constraints that are only supported by the relational data model for LLM benchmarking.
>
> While KGs and RDBs can both store relational data, they are based on different data models. RDBs are based on the relational data model and assume a fixed schema, which enables strong data integrity based on database design theory. KGs are based on the graph data model and have a schema-less design, which means the format is more flexible, but it may be more challenging to maintain the data integrity.
>
> The key idea of ERBench is to utilize data integrity constraints in RDBs for better LLM benchmarking. In particular, ERBench uses functional dependencies (FDs) to automatically pinpoint critical keywords. Also, ERBench constructs multi-hop questions in a more principled way by joining relations with foreign key constraints (FKCs). FDs and FKCs are universally used in databases, so any database can be used to construct an LLM benchmark using ERBench.
>
> Existing KG-based methods cannot easily support automatic rationale verification and multi-hop question generation. Without FDs, one would have to use some custom method to determine what keywords or phrases to look for in the rationales to verify them. As a result, most KG-based evaluation methods only verify the answers, but not the rationales [1, 2, 3]. Without FKCs, one cannot easily construct arbitrarily-long multi-hop questions that involve multiple entities that are connected. Existing KG-based multi-hop methods [4, 5] require manual curation of bridge entities [4] or logical rules [5] to make the connections.
>
> We thus believe ERBench is the first to utilize RDBs for LLM evaluation with unique benefits and complements KG-based approaches.
>
> &nbsp;
>
> [1] Sun, Kai, et al. "Head-to-tail: How knowledgeable are large language models (llm)? AKA will llms replace knowledge graphs?." NAACL 2024.
>
> [2] Feng, Chao, Xinyu Zhang, and Zichu Fei. "Knowledge solver: Teaching llms to search for domain knowledge from knowledge graphs." ArXiv 2023.
>
> [3] Qian, Cheng, Xinran Zhao, and Sherry Tongshuang Wu. ""Merge Conflicts!" Exploring the Impacts of External Distractors to Parametric Knowledge Graphs." ArXiv 2023.
>
> [4] Yang, Zhilin, et al. "HotpotQA: A dataset for diverse, explainable multi-hop question answering." EMNLP 2018.
>
> [5] Ho, Xanh, et al. "Constructing a multi-hop QA dataset for comprehensive evaluation of reasoning steps." COLING 2020.
>
>
>
> &nbsp;
>
> > **Comment 2. Knowledge coverage with datasets**
>
> Thanks, we clarify that 1) a dataset size of 1K-2K is commonly used in the LLM evaluation literature [1, 6, 7] and 2) the current dataset sizes are sufficient to demonstrate hallucinations of the LLMs we used. Our experiments already show wide ranges of hallucination rates ($H \in [0,1]$) [1] for the six LLMs: 0.15 to 0.95 for the Movie dataset and 0.17 to 0.98 for the Soccer dataset in Table 2. In case the LLMs hallucinate less, we can easily construct ERBench on larger databases to better stress test the LLMs.
>
> &nbsp;
>
> [6] Zhao, Yiran, et al. "Felm: Benchmarking factuality evaluation of large language models." NeurIPS 2024.
>
> [7] Zhang, Muru, et al. "How language model hallucinations can snowball." ArXiv 2023.
>
>
> &nbsp;
> > **Comment 3. Definition of known entity**
>
> Thanks, we meant that the LLM can say something about the entity and its attributes, even if the answer is mechanical. We did not mean the LLM has a true understanding of the entity in a philosophical sense.
>
> &nbsp;
>
> > **Comment 4. New findings**
>
> Thanks, we believe there are new findings that certainly go beyond simple observations and explain as follows:
>
> - Rationale performance analysis
>
> As we mentioned in Comment 1, ERBench supports *automatic rationale verification*.
> Since we are now evaluating rationale in addition to just answers, we can get more detailed insights on how an LLM is performing using the metrics A (answer accuracy), R (rationale accuracy), AR (answer & rationale accuracy), and H (hallucination rate). *In our experiments*, Mistral shows higher answer accuracy (54%) than GPT-4 (47%), but much lower answer-rationale accuracy (6% vs GPT-4’s 41%) on the Soccer dataset.
>
> &nbsp;
>
> - Fine-grained performance analysis
>
> ERBench supports *multi-hop question* generation, which enables fine-grained performance analysis. One can automatically generate multi-hop questions by simply joining multiple relations using FKCs. The number of hops can increase arbitrarily. Since the FDs are preserved after joining tables, the rationale evaluation can be done in a stepwise fashion where we can see how the correctness at one hop impacts the correctness at the next hop. *In our experiments*, we show that the first hop accuracy largely matches the final accuracy (Table 4), and in rare cases an incorrect hop may be followed by a correct hop (Table 16).
>
> &nbsp;
>
> In general, ERBench opens up many opportunities to perform such new analyses.

---

### Official Review · Reviewer_2QMS · 2024-08-03
**Difference compared to existing KG-based complex QA datasets**

**Rating:** 6
**Confidence:** 3
**Correctness:** I believe the claims made and the res…
**Clarity:** The paper is well written and easy to…

**Review:**

The paper provides a thorough evaluation of multiple LLMs giving insight into hallucination rates of various models. I have a few concerns regarding the novelty of the proposed dataset:

1. The paper proposes methods for creating various complex question-answer set from knowledge graphs. However, there exists various datasets in literature that were generated using KGs for complex question answering, for example Lc-quad [1]. I believe the method for creating questions in LC-quad is similar to this papers work (except using LLM for automatic question generation). Therefore, it would be good to discuss the differences between this dataset and the older kg-based QA datasets like LC-quad and why we cannot use these existing datasets for evaluating factual hallucinations in LLMs.

[1] Dubey, M., Banerjee, D., Abdelkawi, A. and Lehmann, J., 2019. Lc-quad 2.0: A large dataset for complex question answering over wikidata and dbpedia. In The Semantic Web–ISWC 2019: 18th International Semantic Web Conference.

2. There also exists multi hop datasets like hotpotqa and 2wikimultihop based on relationships between multiple entities in Wikipedia. It would be useful to describe why such datasets are inadequate to measure factual hallucinations in multi-hop setting.

3. I believe the intermediate rationale verification is very simple because it only check for presence of keywords in the response and not not how it was used in context.

**Strengths:**

The main strength of the paper is the detailed evaluation of multiple LLMs on various types of proposed question answer samples that are generated from any generic knowledge graph. Additionally, this method is scalable to any KG and therefore is a generic method to create more samples as there are updates to the entities/new entities in the KG.

**Additional Feedback:**

No additional feedback. Please see the concerns raised above.

**Documentation:**

Yes the paper provides detailed description on how the dataset was created and the how the evaluations of various LLM were conducted.

**Ethics:**

No ethical concerns.

**Limitations:**

Yes the authors have addressed the limitations.

**Opportunities For Improvement:**

Please see the concern raised above.

**Relation To Prior Work:**

Yes prior work is discussed in relation to datasets used for benchmarking LLM performance. However, it would be useful to discuss about the comparison with older KG-based complex QA datasets also.

**Summary And Contributions:**

The paper proposes a dataset for measuring factual hallucinations in Large Language Models (LLMs) by creating various types of question answering samples like binary, multiple choice, and multi-hop questions from knowledge bases. The paper provides a thorough evaluation of various LLMs on the proposed dataset which is impressive.

---

> ### Author Rebuttal · Authors · 2024-08-16
>
> We appreciate your thoughtful comments and useful suggestions. We are happy to address them below and in our revision.
>
> &nbsp;
>
> > **Comment 1. Comparison with KG-based QA construction**
>
> Thanks, we clarify that ERBench is not based on knowledge graphs (KGs), but on relational databases (RDBs), which have fundamental differences. While both KGs and RDBs can store data, they are based on different data models. RDBs are based on the relational data model and assume a fixed schema, which enables strong data integrity based on database design theory. KGs are based on the graph data model and have a schema-less design, which means the format is more flexible, but it may be more challenging to maintain the data integrity.
>
> The key idea of ERBench is to utilize data integrity constraints in RDBs for better LLM benchmarking. In particular, ERBench uses functional dependencies (FDs) to automatically pinpoint critical keywords. Also, ERBench constructs multi-hop questions in a more principled way by joining relations with foreign key constraints (FKCs). FDs and FKCs are universally used in databases, so any database can be used to construct an LLM benchmark using ERBench.
>
> Existing KG-based methods like LC-quad cannot easily support automatic rationale verification and multi-hop question generation. Without FDs, one would have to use some custom method to determine what keywords or phrases to look for in the rationales to verify them. As a result, most KG-based evaluation methods only verify the answers, but not the rationales [1, 2, 3]. Without FKCs, one cannot easily construct arbitrarily-long multi-hop questions that involve multiple entities that are connected.
>
> We thus believe ERBench is the first to utilize RDBs for LLM evaluation with unique benefits and complements KG-based approaches.
>
> &nbsp;
>
> [1] Sun, Kai, et al. "Head-to-tail: How knowledgeable are large language models (llm)? AKA will llms replace knowledge graphs?." NAACL 2024.
>
> [2] Feng, Chao, Xinyu Zhang, and Zichu Fei. "Knowledge solver: Teaching llms to search for domain knowledge from knowledge graphs." ArXiv 2023.
>
> [3] Qian, Cheng, Xinran Zhao, and Sherry Tongshuang Wu. ""Merge Conflicts!" Exploring the Impacts of External Distractors to Parametric Knowledge Graphs." ArXiv 2023.
>
> &nbsp;
>
> > **Comment 2. Comparison with other multi-hop question generation**
>
> We compare HotPotQA and 2WikiMultiHopQA with ERBench in terms of (1) what they evaluate and (2) how they are constructed and show how the two approaches are complementing.
>
> First, there is a clear difference in what is being evaluated. For HotPotQA and 2WikiMultiHopQA, an LLM is given paragraphs along with questions about them, so the goal is to assess the natural language understanding capability of the LLM. In comparison, ERBench does not provide any paragraph and focuses on evaluating the pre-trained knowledge of the LLM.
>
> Second, the benchmark construction is also different. ERBench automatically generates multi-hop questions by simply joining multiple relations using FKCs. The number of hops can increase arbitrarily. Since the FDs are preserved after joining tables, the rationale evaluation can be done in a stepwise fashion where we can see how the correctness at one hop impacts the correctness at the next hop. In comparison, both the HotPotQA and 2WikiMultiHopQA papers go through relatively more complex/expensive steps to generate multi-hop questions. For HotPotQA, the authors 1) build a hyperlink graph and retrieve a “bridge entity” that connects the two entities based on a set of manually created pages or 2) manually curate lists of similar entities and  randomly sample two paragraphs. Then the authors use crowdsourcing to generate question and answer pairs. For 2WikiMultiHopQA, the authors  randomly choose two entities, obtain all triples of relations and objects for the entities, obtain a set of mutual relations between the two entities, obtain wikipedia information, and test whether the collected data passes the predefined requirements.
>
> While we do not claim that ERBench’s approach is better than the others, it is effective in generating multi-hop questions based on RDBs.
>
> &nbsp;
>
> > **Comment 3.  Simplicity of keyword based verification / not considering the context**
>
> ERBench’s keyword-based verification is effective as it relies on powerful database design theory where it pinpoints important keywords that must appear in an LLM’s rationale based on FDs regardless of the context. We did perform experiments on how correct the verification is in Sections 4.2 and B.4. As a result, ERBench has higher than 94% correctness on average using either human evaluation or GPT4-Judge.
>
> To also use context information, we can use extended notions of FDs like conditional functional dependencies (CFDs) [4], which are FDs that hold only under certain circumstances (e.g., FDs that only hold in a certain country). We believe CFDs can also be used to attach context to keywords for more comprehensive verification and may be a solution to address the incorrect 6% above.
>
> &nbsp;
>
> [4] Bohannon, Philip, et al. "Conditional functional dependencies for data cleaning." ICDE 2007.

---

### Author Rebuttal · Authors · 2024-08-17

> **General response**

We appreciate your thorough reviews and valuable suggestions. We would like to emphasize that our method ERBench utilizes relational databases (RDBs), which are based on a different data model compared to knowledge graphs (KGs). RDBs are based on the *relational data model* and assume a *fixed schema*, which enables strong data integrity. In contrast, KGs are based on the *graph data model* and assume a *schema-less design*, which gives more flexibility, but has weaker data integrity.

ERBench is the first to utilize RDBs for LLM evaluation and provides different benefits from existing KG-based approaches (automatic rational verification and systematic multi-hop question generation), leveraging data integrity constraints of RDBs. We will provide more detailed comparisons in response to related comments.

---

### Decision · Program_Chairs · 2024-09-26

**Decision:**

Accept (Spotlight)

**Comment:**

The paper proposes a new method relying on integrity constraints of databases to convert any database into an LLM benchmark. This is an interesting idea that differs from the studies using knowledge graphs. ERBench can verify the corectness of the answer as well as its rationale.
The contribution is new and we hope it will inspire others to do more research in this direction. For this reason we recommend accepting it as a spotlight poster.